# Nutritional state-dependent modulation of insulin-producing cells in *Drosophila*

Rituja S Bisen[1], Fathima Mukthar Iqbal[1], Federico Cascino-Milani[1], Till Bockemühl[2], Jan M Ache[1]*

[1]Neurobiology and Genetics, Theodor-Boveri-Institute, Biocenter, Julius-Maximilians-University of Würzburg, Würzburg, Germany; [2]Department of Animal Physiology, Institute of Zoology, University of Cologne, Cologne, Germany

## eLife Assessment

With **compelling** electrophysiological and behavioural evidence, this work establishes that the activity of insulin-producing cells (IPCs) depends on the nutritional state in *Drosophila* and that, like in mammals, there is also an incretin-like effect with IPCs responding to glucose feeding but not to glucose perfusion. Moreover, the authors demonstrate that DH44 neurons respond to glucose perfusion and, together with IPCs, modulate locomotor activity. This **important** study on the neuronal regulation of metabolic homeostasis will be of interest to both neuroscience and to medical research in diabetes.

**\*For correspondence:**
jan.ache@uni-wuerzburg.de

**Competing interest:** The authors declare that no competing interests exist.

## Abstract

Insulin plays a key role in metabolic homeostasis. *Drosophila* insulin-producing cells (IPCs) are functional analogues of mammalian pancreatic beta cells and release insulin directly into circulation. To investigate the in vivo dynamics of IPC activity, we quantified the effects of nutritional and internal state changes on IPCs using electrophysiological recordings. We found that the nutritional state strongly modulates IPC activity. IPC activity decreased with increasing periods of starvation. Refeeding flies with glucose or fructose, two nutritive sugars, significantly increased IPC activity, whereas non-nutritive sugars had no effect. In contrast to feeding, glucose perfusion did not affect IPC activity. This was reminiscent of the mammalian incretin effect, where glucose ingestion drives higher insulin release than intravenous application. Contrary to IPCs, Diuretic hormone 44-expressing neurons in the pars intercerebralis (DH44^PINs) responded to glucose perfusion. Functional connectivity experiments demonstrated that these DH44^PINs do not affect IPC activity, while other DH44Ns inhibit them. Hence, populations of autonomously and systemically sugar-sensing neurons work in parallel to maintain metabolic homeostasis. Accordingly, activating IPCs had a small, satiety-like effect on food-searching behavior and reduced starvation-induced hyperactivity, whereas activating DH44Ns strongly increased hyperactivity. Taken together, we demonstrate that IPCs and DH44Ns are an integral part of a modulatory network that orchestrates glucose homeostasis and adaptive behavior in response to shifts in the metabolic state.

## Introduction

Insulin is a key neuropeptide that controls metabolic homeostasis across the animal kingdom (*Barbieri et al., 2003*; *Brogiolo et al., 2001*; *Garofalo, 2002*; *Jin Chan and Steiner, 2000*). Beyond metabolic homeostasis, insulin signaling is implicated in physiological processes underlying reproduction, aging, and stress resistance (*Broughton et al., 2005*; *Partridge et al., 2011*; *López-Otín et al., 2013*). When nutrients are abundant, insulin promotes the uptake and storage of energy. Conversely, during periods of starvation, insulin release is downregulated (*Cahill, 2006*; *Abdul-Ghani et al., 2006*),

**eLife digest** Most animals, including humans, use a hormone called insulin to regulate their blood sugar levels and the balance of energy in their bodies. In fruit flies, insulin is released into the hemolymph – the liquid that acts as blood in insects – by neurons that sit directly on top of the brain. These 'insulin-producing cells', or IPCs, play the same role as the beta cells that create insulin in the human pancreas. As fruit flies can easily be genetically manipulated, IPCs have drawn increased interest as a model system to investigate insulin regulation. However, exactly how these cells behave in live insects has remained poorly understood.

To address this knowledge gap, Bisen et al. used an approach that allowed them to record the activity of individual IPCs in live fruit flies under different conditions. The results showed that IPCs release insulin when sugars are eaten as part of a meal, but not when these molecules are directly introduced into the insects via injection. In humans, this well-known phenomenon is known as the incretin effect; it points to insulin release being controlled by complex mechanisms involving gut hormones from the digestive system rather than by a simple increase in blood sugar levels. Bisen et al. also found that the activity of IPCs was much lower in older flies, which may indicate changes in how the insects process sugars later in life.

Whether insects 'decide' to look for food is closely tied to variations in their energy reserves, which are linked to insulin release. Additional experiments during which IPCs were artificially stimulated (therefore replicating what would normally happen following a meal and an increase in circulating sugar levels) showed that these cells only play a minor role in modulating food searching behavior compared to other neurons such as DH44-producing cells.

Taken together, these results refine our understanding of the circuits that control insulin release in fruit flies, allowing further examinations that could lead to insights relevant to human health and diseases, such as diabetes.

which promotes the conservation of energy resources and enables energy mobilization via counter-regulatory pathways (*Carlson et al., 1994*; *McCue, 2010*; *Finn and Dice, 2006*). Maintaining this energy balance is vital for survival. Accordingly, dysregulation of this finely tuned system is a hallmark of metabolic disorders such as type 2 diabetes (*DeFronzo, 1988*; *Hull et al., 2004*; *Kahn, 2003*).

*Drosophila melanogaster* has emerged as a model system to study insulin signaling because of the availability of a powerful genetic toolkit (*Bellen et al., 2010*; *Baker and Thummel, 2007*; *Morris et al., 2012*; *Chatterjee and Perrimon, 2021*; *Park et al., 2014*; *Jennings, 2011*). In *Drosophila*, 14 IPCs, which are analogous to human pancreatic beta cells, reside in the pars intercerebralis (PI) of the brain (*Nässel and Winther, 2010*). These IPCs secrete *Drosophila* insulin-like peptides (DILPs) and are sensitive to changes in the metabolic state (*Fridell et al., 2009*; *Kim and Neufeld, 2015*; *Barber et al., 2016*; *Géminard et al., 2009*), the behavioral state (*Liessem et al., 2023*), the circadian time (*Barber et al., 2016*; *Barber et al., 2021*), aminergic and peptidergic modulatory inputs (*Luo et al., 2012*; *Held et al., 2024*), and other peripheral signals (*Nässel and Vanden Broeck, 2016*). Consequently, they play a key role in orchestrating the metabolic demands of behaving animals. For instance, insulin acts as a global cue for modulating odor-driven food-search in accordance with the fly's nutritional state (*Root et al., 2011*). Unlike pancreatic beta cells, IPCs are localized in the brain and engage in direct synaptic signaling (*Held et al., 2024*; *Nässel et al., 2013*) in addition to endocrine signaling. Thus, IPCs are part of a complex system that integrates both neuronal and peripheral signals. As a key hub integrating metabolic state information, IPCs are thought to be part of the neuronal circuitry that regulates food intake (*Root et al., 2011*; *Marella et al., 2012*; *Pathak and Varghese, 2021*; *Ribeiro and Dickson, 2010*; *Brent and Rajan, 2020*; *Yang et al., 2015*; *Jourjine et al., 2016*), in part by regulating foraging behavior. Previous studies found that octopamine, the insect analogue to norepinephrine, mediates starvation-induced hyperactivity in flies (*Yang et al., 2015*). A subset of octopaminergic neurons expressing *Drosophila* insulin receptor (dInR) are necessary for the regulation of starvation-induced hyperactivity (*Yang et al., 2015*; *Yu et al., 2016*) implicating that IPCs play a role in modulating foraging behavior.

While much is known about carbohydrate metabolism and insulin signaling in *Drosophila* (*Broughton et al., 2005*; *Enell et al., 2010*; *Kapan et al., 2012*; *Luo et al., 2014*), the in vivo dynamics

of IPCs remain largely elusive. Ex vivo studies indicate that IPCs themselves are sensitive to glucose, as evidenced by increased firing rates upon glucose perfusion (*Fridell et al., 2009*; *Kréneisz et al., 2010*). However, the influence of feeding and nutritional state changes on IPC activity in vivo, as well as the nutritional state-dependent response of IPCs to glucose, remain unclear, underscoring the importance of in vivo studies where all nutrient-sensing and sensorimotor circuits are intact. For instance, how does IPC activity differ when glucose is ingested as part of a meal, compared to when glucose is administered directly to the brain or into circulation? In mammalian systems, insulin release is significantly amplified upon glucose ingestion vs. intravenous administration (*Elrick et al., 1964*; *DeFronzo et al., 1978*), a phenomenon known as the 'incretin effect' (*Creutzfeldt, 1979*; *Kazafeos, 2011*). This underlines the complexity of insulin regulation, emphasizing that insulin secretion is not solely dictated by sugar levels but influenced by multiple factors. Here, we investigated the nutritional state-dependent modulation of IPCs in *Drosophila* using an in vivo electrophysiological approach.

We recently demonstrated that IPC activity is strongly modulated by locomotion on fast timescales (*Liessem et al., 2023*). Whether the converse is true, and IPC activity can directly affect behavior on short timescales, is not clear. To address this, we optogenetically manipulated IPCs and other modulatory neurons while recording the locomotor activity of freely walking flies.

In combination, our approaches show that IPC activity is not just nutritional state-dependent, but that IPCs also exhibit an incretin-like effect. We further establish that the nutritional state strongly modulates locomotor activity, and that IPCs play a small but significant role in this. In addition, we confirm that another set of modulatory neurons adjacent to the IPCs, DH44[PI]Ns, are sensitive to glucose perfusion. While DH44[PI]Ns do not affect IPC activity on short timescales, DH44 neurons outside the PI (DH44Ns) form strong inhibitory connections to IPCs. These DH44Ns also have strong effects on locomotor activity, which are antagonistic to those of IPCs. Our findings imply distinct mechanisms by which IPCs and DH44[PI]Ns sense changes in nutritional state, underlining the significance of multiple pathways in orchestrating *Drosophila*'s response to nutritional state changes and ensuring metabolic homeostasis.

## Results

### IPC activity is modulated by starvation and glucose-feeding

To characterize the electrophysiological activity of IPCs in different nutritional states, we performed in vivo patch-clamp recordings from IPCs in fed and starved *Drosophila* (*Figure 1A–C*). First, flies were fed ad libitum, and the IPC baseline activity was determined by recording their spontaneous activity in glucose-free extracellular saline in a 5 min window. The baseline firing rate varied between individual IPCs and ranged from 0 to 1.4 Hz (*Figure 1D*). To assess the effect of starvation on IPC activity, we wet-starved flies for 24 h. IPCs were significantly less active in starved flies than in fed flies with a median firing rate of 0 Hz (*Figure 1D*). Hence, the IPC population was basically quiescent after starvation. We also analyzed the resting membrane potential ($V_m$) of IPCs in fed and starved flies. On average, the $V_m$ of IPCs was 8 mV more depolarized (median: –51 mV) in fed flies compared to starved flies (median: –59 mV, *Figure 1E*). This indicates a higher excitability of IPCs after feeding, which likely explains their higher firing rates.

To explore the temporal dynamics of starvation-induced decreases in IPC activity, we recorded IPCs in flies subjected to increasing durations of starvation. The IPC population activity decreased steadily with increasing starvation durations and reached a median of 0 Hz after 24 h (*Figure 1G*). The fraction of active IPCs, that is, IPCs that fired at least one action potential during our five min recording window, also progressively decreased with increasing periods of starvation (*Figure 1G*, gray). The reduction in IPC activity was already noticeable after 0.5–2 h of starvation, suggesting that starvation tightly regulates IPC activity and hence insulin release on shorter and longer timescales.

Having established that starvation reduces IPC activity, we next explored if IPC activity could be restored by feeding. Therefore, we first starved flies for 24 h and then refed them with a high glucose (HG) diet (400 mM glucose in 2% agar) for durations ranging from 3 to 24 h (see *Figure 1F and G*). IPC activity in flies that were refed for 3–5 h resembled that of starved flies and remained around 0 Hz (*Figure 1G*). 6 h after the start of refeeding, IPC activity and the fraction of active IPCs started to increase and were similar to that of fed flies (*Figure 1G*, p=0.68), indicating that this duration was sufficient to allow IPC activity to fully recover from starvation. Interestingly, IPC activity kept increasing

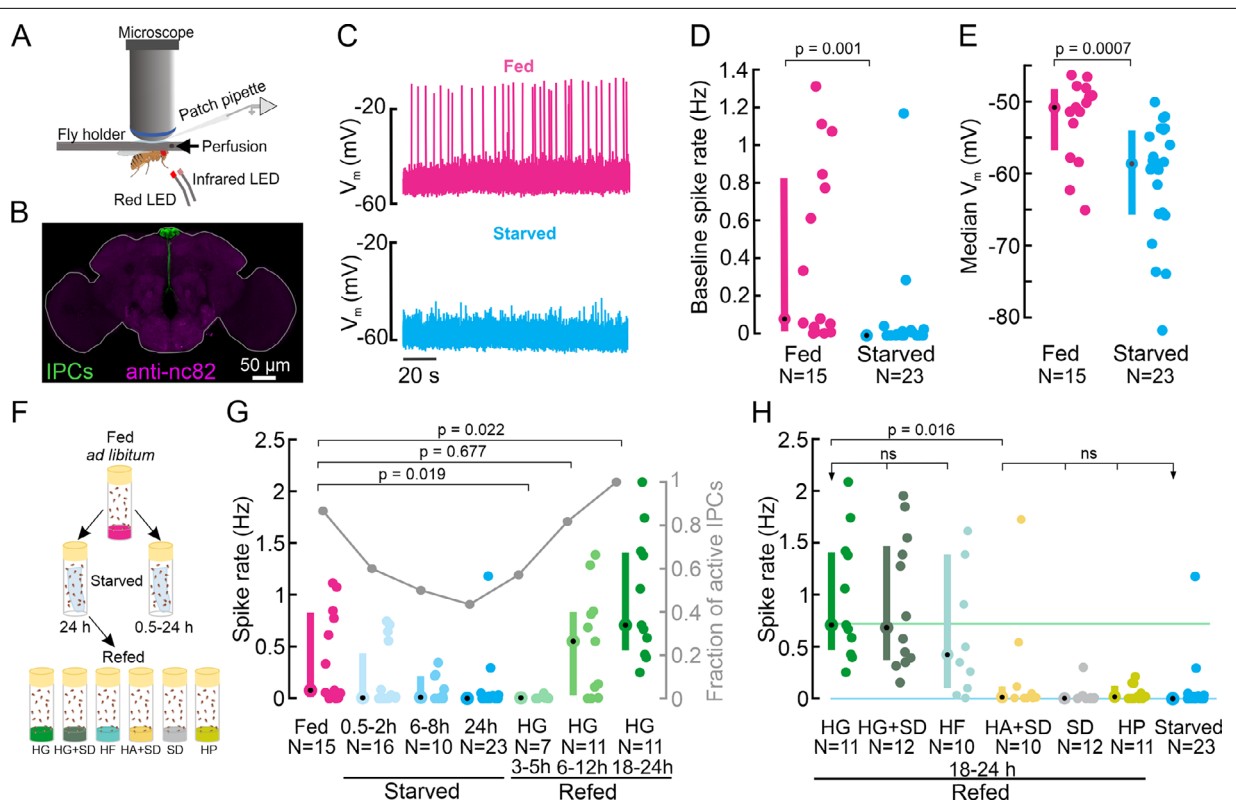

**Figure 1.** Insulin-producing cell (IPC) activity is modulated by the nutritional state and increases in response to feeding on nutritive sugars. (**A**) Schematic of the setup for in vivo IPC whole-cell patch-clamp recordings. (**B**) IPCs in the *Drosophila* brain. *UAS-myr-GFP* was expressed under a *Dilp2-GAL4* driver to label all 14 IPCs. The GFP signal was enhanced with anti-GFP (green), and brain neuropils were stained with anti-nc82 (magenta). (**C**) Representative examples of the membrane potential of two IPCs recorded in fed (magenta) and starved (cyan) flies. (**D**) Average baseline spike rate and (**E**) membrane potential of IPCs in fed (magenta) and starved (cyan) flies. (**F**) Schematic of the experimental starvation and refeeding protocol. HG: High glucose, HG + SD: High glucose with a standard diet, HF: High fructose, HA + SD: High arabinose with a standard diet, SD: Standard diet, HP: High protein. (**G**) Comparison of IPC spike rates in fed flies (magenta), increasingly starved flies (cyan), and flies refed on HG for different durations (green). Right axis shows a fraction of active IPCs (number of IPCs with spike rate >0 Hz, gray lines, and circles). (**H**) Comparison of IPC spike rate in flies refed on different diets. Reference lines represent median IPC activity in flies refed with HG for 18–24 h (green), and in flies starved for 24 h (cyan). Each circle represents an individual IPC, N=number of IPCs (see *Supplementary file 1e* for a number of flies), error bars indicate the median (circle) and inter-quartile range (IQR, bars). For D, E, and G, p-values are reported from the Wilcoxon rank-sum test. For H, a Kruskal-Wallis test followed by post hoc Wilcoxon rank-sum tests were used for pairwise comparisons. p-values are reported after the Holm-Bonferroni correction.

The online version of this article includes the following figure supplement(s) for figure 1:

**Figure supplement 1.** Dietary restriction impairs survival in *Drosophila*.

**Figure supplement 2.** Sex and mating state do not affect insulin-producing cell (IPC) activity, but aging does.

for longer glucose feeding durations, and was significantly increased in flies that were refed with glucose for 18–24 h compared to flies kept on a standard diet and fed ad libitum (*Figure 1G*, p=0.02). Together, these results clearly show that IPC activity decreases with increasing duration of starvation up to 24 h, and slowly but steadily recovers and eventually overshoots the baseline after refeeding with glucose.

To ensure that the results obtained on an HG diet were not due to a lasting interference with the fly's metabolic system, we conducted a survival assay. Analogous to our electrophysiology experiments, we starved flies for 24 h and refed them on HG. Subsequently, we monitored the number of survivors every day while flipping flies onto fresh food every second day. As expected, flies refed with our standard diet (SD) survived the longest (100% survival until day 43, *Figure 1—figure supplement 1*). 100% survival was observed for flies fed with HG until day 12 and at least half of the flies survived until day 16 (*Figure 1—figure supplement 1*). Since our IPC recordings were acquired in flies refed

for a maximum of 24 h, the effects we observed on IPC activity were not due to lethality in flies but rather due to differences in nutrient availability.

## IPCs are sensitive to the ingestion of nutritive sugars

The survival assay indicated that the HG diet was stressful for flies, as it did not contain all the necessary nutrients. To ensure that the increased IPC activity was not a stress response but due to glucose ingestion, we added 400 mM glucose to our standard diet (HG + SD) and reassessed the effect of glucose ingestion in a full diet background. Refeeding flies with HG + SD caused a significant increase in IPC activity (*Figure 1H*), which was indistinguishable from that of HG only (*Figure 1H*), confirming that IPCs respond to glucose feeding.

Next, we tested whether IPC responses are glucose-specific – in principle, IPCs could also be sensitive to the ingestion of other nutritive sugars. To this end, we kept flies on a high fructose diet (HF, 400 mM Fructose in 2% agar), which is a natural source of nutritive sugar for flies, found in fruit. IPCs in flies kept on the HF diet exhibited a similar activity level as in flies kept on HG (*Figure 1H*), suggesting IPCs generally respond to the ingestion of nutritive sugars. To confirm this hypothesis, we investigated if IPCs respond to a non-nutritive but sweet-tasting sugar. Hungry flies are motivated to feed on non-nutritive sugars, and the sweet taste or sugar concentrations may affect insulin signaling. To test this, we refed flies with 400 mM D-arabinose. Since arabinose is non-nutritive, we supplied it in our standard diet (HA + SD). In contrast to nutritive sugars, refeeding flies with D-arabinose did not increase IPC activity after starvation. Accordingly, IPC activity was significantly lower than in flies fed with HG or HF, and indistinguishable from starved flies (*Figure 1H*). These results clearly show that IPC activity is specifically increased in response to the ingestion of nutritive sugars.

It was unexpected that IPC activity after refeeding with the HA + SD diet was comparable to starved flies, since we expected it to recover to levels observed in flies fed ad libitum (*Figure 1G*, magenta). However, refeeding flies on SD only, without additional sugars, did not increase IPC activity compared to the 24 h starved state, even though we refed flies for 18–24 h, where we saw the maximal effect of glucose refeeding (*Figure 1H*). This was likely because sugar concentrations in our SD were low (see Discussion). To further confirm whether IPC activity increases after refeeding were sugar-specific or could also be recovered by other specific nutrients, like protein, we kept 24 h starved flies on a high protein diet (HP) and compared the IPC activity after refeeding for 18–24 h. Similar to refeeding on SD, refeeding on an HP diet caused no significant increase in the spike rate of IPCs (*Figure 1H*). This was intriguing, because both protein-rich diets and standard diets have been shown to induce insulin secretion under fed conditions (*Géminard et al., 2009*; *Söderberg et al., 2012*; *Agrawal et al., 2016*). In comparison to HG and SD, flies survived poorly on the HP diet: 100% survival was observed only up to day one, and only 43% of the flies survived longer than day 12 (*Figure 1—figure supplement 1*). We can, therefore, not rule out the possibility that flies on HP exhibited a significantly impaired metabolism. Overall, our refeeding experiments demonstrate that IPCs specifically respond to feeding on nutritive sugars and exhibit relatively slow temporal dynamics on the scale of hours in this context.

Since IPCs are central to metabolic regulation and development, factors other than the nutritional state may impact their activity. To address two key factors, we analyzed whether IPC activity was affected by the mating state or the age of the fly. Although median IPC activity was the highest in males, there were no significant differences between virgin females, mated females, and male flies (*Figure 1—figure supplement 2A*). Hence, IPC activity is not significantly affected by sex or female mating state. However, when recording the IPC activity in females of different age groups, we observed that IPCs remained almost quiescent in fully fed females older than 10 d. Here, baseline spike rates ranged from 0 to 0.4 Hz and were significantly lower than in younger females (*Figure 1—figure supplement 2B*). This indicates that IPC activity is reduced in older flies. Therefore, aging significantly reduces IPC activity.

## The nutritional state affects locomotor activity

Starvation increases baseline locomotor activity in flies and other animals. This 'starvation-induced hyperactivity' is a hallmark of foraging behavior (*Root et al., 2011*; *Yang et al., 2015*; *Yu et al., 2016*; *Lee and Park, 2004*; *Krashes et al., 2009*; *Tsao et al., 2018*; *Geo et al., 2019*) as it enables animals to explore their environment and locate food sources. As major integrators of metabolic state

information, IPCs are thought to be part of the neuronal circuitry that modulates foraging (*Yu et al., 2016*). For example, insulin inhibits a subset of olfactory sensory neurons, such that decreased insulin release leads to an increase in sensitivity to food odors (*Ko et al., 2015*). Our findings that IPC activity is directly modulated by starvation and feeding support the hypothesis that they could be part of the neuronal networks modulating foraging. To further investigate the relationship between the nutritional state, IPC activity, and foraging behavior, we used our 'Universal Fly Observatory (UFO)' setup to quantify locomotor activity (*Chockley et al., 2022*), in particular starvation-induced hyperactivity (*Figure 2A*). In the UFO, flies can walk freely in a circular arena illuminated by infrared LEDs, and their walking activity can be quantified using semi-automated tracking software and analysis (based on the Caltech Fly Tracker *Eyjolfsdottir et al., 2014*, see Methods). To test whether IPCs play a causal role in modulating starvation-induced hyperactivity, we first quantified the effects of nutritional manipulations on walking speed. Fully fed flies, which have elevated IPC activity, displayed a low baseline walking activity with a median forward velocity (FV) of 0.3 mm/s (*Figure 2B*). 24 h starved flies, in which IPC activity was significantly reduced, were more hyperactive and displayed a significantly increased median FV of 1.9 mm/s (*Figure 2B*). Refeeding flies on a pure glucose diet for 24 h after starvation, which led to a drastic increase in IPC activity (see *Figure 1G*), reduced the hyperactivity by about 50% (median FV = 0.9 mm/s) compared to starved flies (*Figure 2B*). Hence, the hyperactivity was significantly reduced but not abolished, despite the fact that IPC activity was maximal in these flies (*Figure 1G*). One interpretation of the persistent increase in activity after glucose refeeding is that flies are searching for protein sources. Flies refed on a full diet (SD) were even less active than flies fed ad libitum before undergoing starvation, and this difference was significant (*Figure 2B*). These results show that the locomotor activity was affected by the same dietary manipulations that had strong effects on IPC activity. However, IPC activity changes alone cannot explain the modulation of starvation-induced hyperactivity. On the one hand, high-glucose diets which drove the highest activity in IPCs were not sufficient to reduce locomotor activity back to baseline levels. On the other hand, refeeding flies with SD did not revert the effects of starvation on IPC activity (*Figure 1H*), but it was sufficient to reduce the locomotor activity below baseline levels (*Figure 2B*). This suggests that the modulation of starvation-induced hyperactivity is achieved by multiple modulatory systems acting in parallel.

To develop a deeper understanding of the nutritional state-dependent modulation of behavior, we investigated the temporal dynamics of the changes in locomotor activity. Since the nutritional state-dependent modulation of IPC activity occurred on a relatively slow timescale over several hours to days, we tested whether starvation-induced hyperactivity was modulated on a similar timescale. To this end, we quantified the walking activity of flies that were either fed ad libitum or starved for increasing durations, ranging from 30 min to 48 h. As established earlier, flies that were fully fed or merely starved for 30 min had a low FV of 0.3 mm/s (*Figure 2—figure supplement 1A*). However, their walking activity increased with increasing starvation duration up to 20 h, when it reached a maximum at an FV of 3.6 mm/s (*Figure 2—figure supplement 1A*). The FV of flies starved for >20 h decreased again, probably due to a lack of energy reserves. We observed a final peak after 36 h of starvation, which we interpret as a surge in starvation-induced hyperactivity before the metabolic state becomes critical (*Figure 2—figure supplement 1A*). These results demonstrate that the starvation duration has a strong effect on locomotor activity, the modulation of which occurs on a similar timescale as IPC modulation by changes in nutritional state.

## IPC activation slightly reduces starvation-induced hyperactivity

Having confirmed a strong effect of starvation on both IPC activity and locomotor activity, our next goal was to investigate whether changes in IPC activity contribute to the control of starvation-induced hyperactivity. To achieve this, we optogenetically activated IPCs via CsChrimson (*Klapoetke et al., 2014*) in freely walking flies, which were either fed ad libitum or wet-starved for 24 h. In parallel, we used *empty split-GAL4* (*Hampel et al., 2015*) flies crossed to *UAS-CsChrimson* as controls. Controls and experimental flies were simultaneously exposed to the same activation protocol in adjacent UFO setups. This approach allowed us to account for activity changes due to the red light used for optogenetic activation and other factors potentially affecting walking activity, such as small fluctuations in temperature, humidity, or differences in the circadian time. Each optogenetic activation experiment consisted of five activation cycles, in which we activated the neurons for 300 s, followed by an

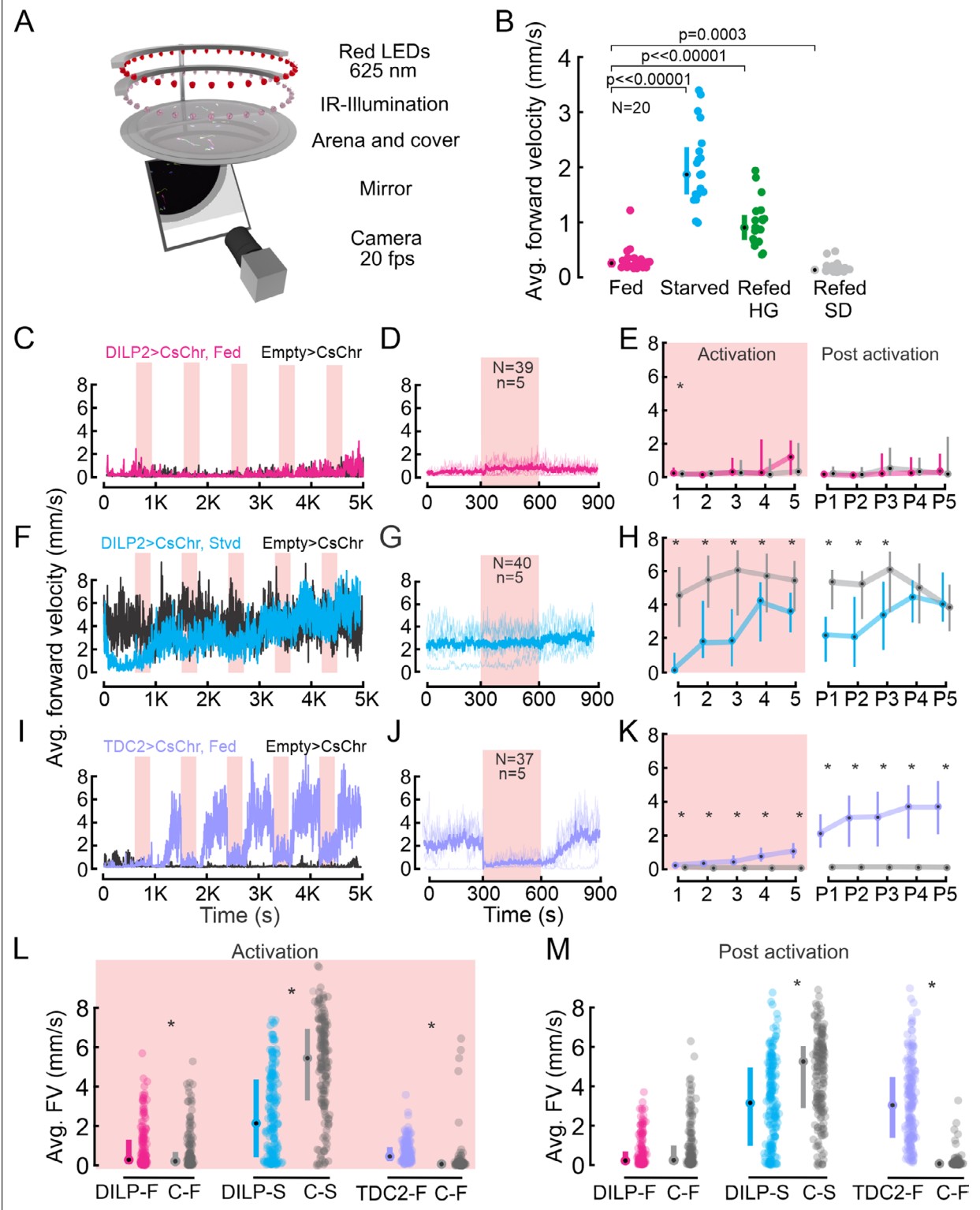

**Figure 2.** Walking activity is modulated by the nutritional state, insulin-producing cells (IPCs), and octopaminergic neurons (OANs). (**A**) Schematic showing the Universal Fly Observatory (UFO) setup. (**B**) Average forward velocity (FV) of flies in different feeding states. Median and IQR are shown. p-values from the Wilcoxon rank-sum test. (**C**) Average FV of flies representing one replicate during optogenetic activation of IPCs in fed flies (magenta). *Empty split-GAL4* was used as the control for all experiments (black). Red shading, optogenetic activation (300 s). (**D**) Average FV across all trials from two replicates for IPC activation in fed flies: 300 s before stimulus onset, during stimulus, and after stimulus offset. N=number of flies, n=number of activation trials. Thin lines represent individual trials, thick lines represent the median of all trials. (**E**) Average FV of all flies during each stimulus trial

*Figure 2 continued on next page*

*Figure 2 continued*

(1-5) and post-stimulus trial 300 s window immediately after activation ceased, (P1–P5) for IPC activation in fed flies. (**F**) Average FV of flies representing one replicate during optogenetic activation of IPCs in starved flies (cyan). (**G**) Average FV across all trials from two replicates for IPC activation in starved flies (plot details as in D). (**H**) Average FV during individual trials for IPC activation in staved flies (plot details as in E). (**I**) Average FV of flies representing one replicate during optogenetic activation of OANs in fed flies (lilac). (**J**) Average of FV across all trials from two replicates for OAN activation in fed flies (plot details as in D). (**K**) Average FV during individual trials for OAN activation in fed flies (plot details as in E). (**L, M**) Average FV pooled across all stimulus (1-5) and post-stimulus trials (P1–P5), respectively. Median and IQR are shown. p-values are reported from the Wilcoxon rank-sum test. Where no detailed p-value is stated, asterisks represent a significant difference. See also *Supplementary file 1a and b*.

The online version of this article includes the following figure supplement(s) for figure 2:

**Figure supplement 1.** Starvation duration and octopaminergic neuron (OAN) activation affect locomotor activity.

inter-stimulus-interval of 600 s for recovery, which we refer to as the 'post activation' window. Results reported here were analyzed by averaging two replicates per experiment.

IPC activation in fed flies had only minor effects on the FV (*Figure 2C–E*). The FV of most flies remained below 2 mm/s throughout the experiment (75th percentile, *Figure 2E*: see 1–5 and P1-P5) and was indistinguishable from the FV of controls during all but the first activation cycle (*Figure 2C and E*: p>0.05). Accordingly, their FV did not differ during most activation windows (median (1-5)=0–0.5 mm/s, *Figure 2E*) or post-activation windows (median (P1-P5)=0–0.1 mm/s). One exception was the first activation cycle, in which the IPC-activated flies walked slightly but significantly faster than the controls (*Figures 2E and I*: p=0.03). When averaging the activity of all flies across all activation windows, there was a small but statistically significant difference of 0.06 mm/s (*Figure 2L*, p=0.01). In the post-activation windows, however, the FV of IPC activated and control flies were indistinguishable (*Figure 2M*). Overall, IPC activation had no notable effects on locomotor activity in fully fed-flies.

Next, we tested the effect of IPCs on starvation-induced hyperactivity by activating IPCs in starved flies. Given that higher IPC activity levels are associated with the fed state, we expected that activating IPCs would reduce starvation-induced hyperactivity. As previously observed, starved flies were generally more active than their fed counterparts and walked with a median FV >2 mm/s (*Figure 2F–H*). The FV of flies during IPC activation remained significantly lower than in control flies over the course of all five activation cycles (*Figure 2F–H*, p<0.05, *Supplementary file 1a*). This confirmed our hypothesis that IPC activation reduces starvation-induced hyperactivity. However, two observations were counterintuitive: First, the FV of IPC-activated flies and control flies converged over time, and there was no significant difference in the control and IPC-activated flies during the last two post-activation windows (*Figure 2F and H*: P4, p=0.23 and P5, p=0.45, see also *Supplementary file 1b*). Hence, while activation of IPCs reduced hyperactivity, this effect did not persist after long-term activation. Second, we noted a decrease in the FV of IPC-activated starved flies even before the first optogenetic activation (*Figure 2F*), possibly due to CsChrimson's sensitivity to visible light (see methods). Overall, these results suggest that IPCs have a small but significant effect on behavior, in that they slightly reduce starvation-induced hyperactivity upon activation.

## Activation of octopaminergic neurons increases locomotor activity

Since changes in IPC activity alone were insufficient to explain starvation-induced hyperactivity, we next investigated octopamine, a neuromodulator known to induce hyperactivity in starved flies (*Yu et al., 2016*). Octopamine is the insect analogue to norepinephrine and is released during flight, a metabolically very demanding behavior in *Drosophila* and other insects (*Brembs et al., 2007*; *Suver et al., 2012*; *Roeder, 2020*). Hence, we expected octopaminergic neurons (OANs) to drive an increase in locomotor activity, rendering them a useful positive control. We used a tyrosine decarboxylase 2 (TDC2) driver line to acutely activate OANs via CsChrimson (*Klapoetke et al., 2014*) in the UFO while tracking the walking activity of flies. TDC2 is required for octopamine synthesis from tyramine (*Monastirioti et al., 1996*) and *TDC2-GAL4* labels all neurons producing octopamine and tyramine. Experiments were carried out following the same approach as for IPCs, and we again ran empty controls in parallel to OAN activation. We observed a strong, significant increase in FV from 0.23 mm/s (median) before OAN activation to 3.68 mm/s after five activation cycles, with a strong FV increase after each activation (*Figure 2I and K*, p<0.001, see *Supplementary file 1a and b*). Notably, the final FV reached after five cycles of OAN activation was comparable to that after 24 h of starvation (compare *Figure 2I* to starved controls in *Figure 2F*). This suggests that OAN activity can explain a large proportion of

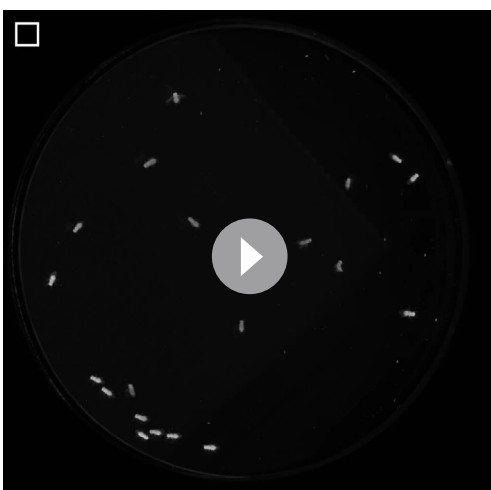

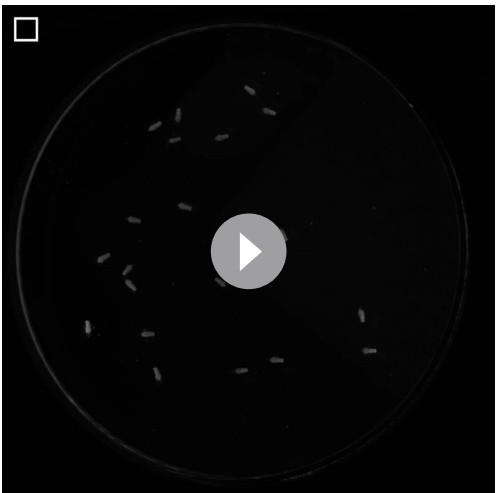

**Video 1.** Behavioral effects of optogenetic octopaminergic neuron (OAN) activation on *Drosophila* locomotor activity in the Universal Fly Observatory (UFO). Example video of OAN-activated flies walking in the UFO during the fifth activation cycle. The white box indicates when the optogenetic stimulation LED is on. The video includes one-minute before activation (P4 in *Figure 2K*), 5 min after activation (5 in *Figure 2K*), and 1-minute after activation (P5 in *Figure 2K*). During OAN activation, the forward velocity decreased significantly, including several halting episodes.
https://elifesciences.org/articles/98514/figures#video1

**Video 2.** Behavioral effects of the optogenetic activation protocol on the locomotor activity of Empty split-Gal4 control flies in the Universal Fly Observatory (UFO). Example video of control flies responding to the light pulse used for optogenetic activation during the fifth activation cycle. These flies were recorded in parallel to the OAN-activated flies in *Video 1*. Details as for *Video 1*.
https://elifesciences.org/articles/98514/figures#video2

the starvation-induced hyperactivity. Interestingly, we observed a dramatic drop in FV at the onset of OAN activation (*Figure 2J and K*), during which the median FV dropped by 2 mm/s and approached 0 mm/s. This reduction in FV was due to recurring stopping behavior observed during acute OAN activation (*Figure 2—figure supplement 1B*, *Videos 1 and 2*). This intermittent locomotor arrest has been previously described in adult flies and is thought to be mediated by ventral unpaired median OANs, which have been suggested to suppress long-distance foraging behavior (*Sayin et al., 2019*). Since these are not the only neurons we activate in the TDC2 line, we speculate that the stopping phenotype could also result from concerted effects of octopamine and tyramine modulating muscle contractions (*Saraswati et al., 2004*; *Selcho et al., 2012*; *Ormerod et al., 2013*) and motor neuron excitability (*Schützler et al., 2019*), as previously described in *Drosophila* larvae, or from OANs interfering with pattern generating networks in the ventral nerve cord (VNC) during longer activation (*Fox et al., 2006*). Overall, long-term optogenetic activation of OANs had a strong and significant effect on walking activity and mimicked the starvation-induced hyperactivity (*Figure 2F and H*, black and gray controls). We conclude that starvation-induced hyperactivity is mildly affected by IPCs, but predominantly controlled by populations of modulatory neurons other than IPCs, including OANs.

## IPCs do not sense changes in extracellular glucose levels in vivo

Our experiments established that IPC activity is modulated by the nutritional state, and that IPCs are sensitive to glucose ingestion, similar to pancreatic beta cells. Like beta cells, IPCs have been suggested to directly sense changes in extracellular glucose concentration in ex vivo studies (*Fridell et al., 2009*; *Kréneisz et al., 2010*), where IPC activity increased in response to the application of saline containing a high glucose concentration to the nervous system. Consequently, IPCs are thought to sense extracellular glucose concentration increases cell-autonomously. To test this hypothesis in flies with fully intact sensory and central circuits, we recorded from IPCs in vivo, while perfusing extracellular saline containing high concentrations of glucose over the brain. After recording the IPC baseline activity in starved flies under the perfusion of glucose-free saline, we perfused the brain with saline containing 40 mM glucose (*Figure 3A*). Similar to our previous experiments, IPC spike rates were very low with a median of 0.05 Hz after starvation. Surprisingly, perfusing high concentrations

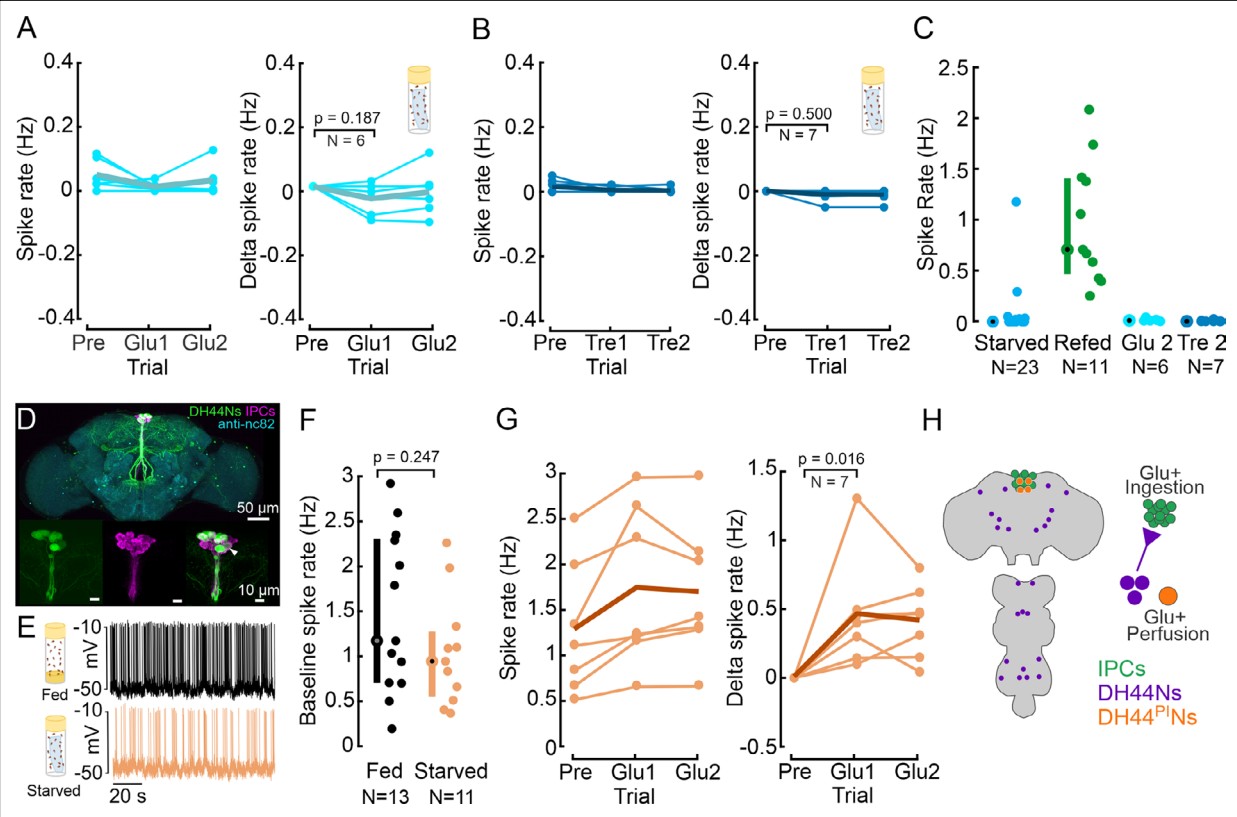

**Figure 3.** Insulin-producing cells (IPCs) are not sensitive to glucose perfusion but DH44^PINs are. (**A**) IPC spike rate and delta spike before and after perfusing extracellular saline with high glucose concentration, in starved flies. Baseline spike rates were recorded in glucose-free saline, followed by recordings in glucose-rich saline. Spike rates were averaged within a 5 min window. Delta spike rate was calculated by subtracting the baseline (Pre) from each trial for each IPC. Pre: 5 min recording in glucose-free saline. Glucose-rich saline was allowed to perfuse for about 8 min before analyzing IPC activity. Glu1 and Glu2: Two subsequent, 5 min long recordings in glucose-rich saline, starting eight minutes after onset of saline perfusion. Each circle represents an individual IPC from a different fly, the thick line represents the grand mean of all recordings. p-values are reported from the Wilcoxon signed-rank test. (**B**) Same experiment as in A, but with perfusion of trehalose-rich saline after recording the baseline. Note that our glucose-free saline was also devoid of trehalose (see methods). Other details same as A. (**C**) Comparison of IPC spike rate between starved, glucose-refed, glucose-perfused, and trehalose-perfused flies, highlighting the 'incretin effect.' Median and IQR are indicated. (**D**) Staining showing *Drosophila* brain with IPCs (magenta) and DH44Ns (green). *UAS-myr-GFP* was expressed under a *DH44-GAL4* driver to label DH44 neurons. GFP was enhanced with anti-GFP (green), brain neuropils were stained with anti-nc82 (cyan), and IPCs were labeled using a Dilp2 antibody (magenta). White arrow indicates Dilp2 and *DH44-GAL4* positive neurons. The other white regions in the image result from an overlap in z-projections between the two channels, rather than from antibody colocalization. (**E**) Example membrane potential of a DH44^PIN recorded in a fed (black) and a starved fly (orange). (**F**) Baseline DH44^PIN spike rate in fed (black) and starved flies (orange). Each circle represents an individual neuron (see ***Supplementary file 1f*** for a number of flies). Median and IQR are indicated. p-values are reported from Wilcoxon rank-sum test. (**G**) Spike rate and delta spike rate of DH44^PINs before and during glucose perfusion in starved flies. Plot details same as A. (**H**) Schematic showing the location of cell bodies (left), and the regulation of IPCs, DH44^PINs, and DH44Ns outside the pars intercerebralis (PI).

The online version of this article includes the following figure supplement(s) for figure 3:

**Figure supplement 1.** Glucose perfusion does not affect the activity of insulin-producing cells (IPCs) and DH44^PINs in fed flies but shifts interspike interval distributions in DH44^PINs.

of glucose over the brain did not increase IPC activity in starved flies (***Figure 3A***), even after 20 min of perfusion. This was similar in fed flies (***Figure 3—figure supplement 1A***). Towards the end of each recording session, we injected current to establish that the IPCs were still excitable, as evidenced by an increase in spike rate.

While IPCs and beta cells canonically sense glucose, the primary circulating sugar in the insect hemolymph is trehalose (***Wyatt and Kalf, 1957***; ***Elbein et al., 2003***), a disaccharide formed by two glucose molecules. Therefore, we tested whether IPCs can sense changes in the extracellular concentration of trehalose. To this end, we repeated our experiments with saline containing an equiosmolar concentration of trehalose (20 mM). As in previous experiments, we recorded the IPC baseline in

glucose-free saline, which was also free of trehalose. Similar to glucose perfusion, we did not observe any significant changes in IPC activity during trehalose perfusion (*Figure 3B*), indicating that IPCs do not sense trehalose either.

These results clearly show that IPCs do not directly sense glucose or trehalose in their extracellular environment in vivo, but rather respond to the ingestion of glucose or other nutritive sugars (*Figure 3C*, see also *Figure 1H*). Hence, IPCs require input from the gut or other sensory pathways, such as taste or mechanosensation, to release insulin in response to glucose ingestion. Taken together, the increase in IPC activity following glucose ingestion but not glucose perfusion resembles the 'incretin effect' observed in mammals, where insulin secretion is significantly higher when glucose is ingested orally compared to isoglycemic intravenous delivery of glucose (*Elrick et al., 1964*; *DeFronzo et al., 1978*). This effect is driven by gut-derived neuropeptides collectively referred to as 'incretin hormones' in mammals (*Creutzfeldt, 1979*; *Kazafeos, 2011*). Recently, incretin-like hormones were also described in *Drosophila* (*Yoshinari et al., 2021*; *Alfa et al., 2015*).

## DH44$^{PI}$Ns sense changes in extracellular glucose levels in vivo

We were surprised to find that IPCs were not responsive to glucose perfusion. To confirm that this was not due to shortcomings of our experimental protocol, we recorded from DH44$^{PI}$Ns. Previous work showed that DH44$^{PI}$Ns are activated by D-glucose via calcium imaging experiments (*Dus et al., 2015*). DH44$^{PI}$Ns are anatomically similar to IPCs, and their cell bodies are located directly adjacent to those of IPCs in the PI, making them an ideal positive control for our experiments (*Figure 3D*). A small subset of DH44$^{PI}$Ns also expresses Dilp2 (*Ohhara et al., 2018*), and our immunostainings confirmed colocalization of Dilp2 and DH44 in a single neuron (*Figure 3D*, white arrow).

When recording from DH44$^{PI}$Ns in vivo, we first observed that their spike amplitude fell within the same range as that of IPCs (*Figure 3E*, ~40–60 mV). However, DH44$^{PI}$N baseline spike rates were higher than those of IPCs, ranging from 0.2 Hz to 3 Hz (*Figure 3F*). Moreover, DH44$^{PI}$Ns exhibited a firing pattern that was more burst-like (*Figure 3E*), with shorter inter-spike-intervals (ISIs) than IPCs (*Figure 3—figure supplement 1B*). When comparing the ISI distribution between DH44$^{PI}$Ns and IPCs, the fraction of ISIs below 500 ms was 60% in DH44$^{PI}$Ns and 19% in IPCs. The cumulative ISI probability distributions differed significantly (*Figure 3—figure supplement 1B*, p<<0.001). Hence, the two neuron types displayed distinct activity patterns.

Next, we tested the effect of starvation on DH44$^{PI}$N activity. The baseline spike rate was slightly lower in starved compared to fed flies, but, unlike in IPCs, this difference was not significant (*Figure 3F*). However, the cumulative probability distribution of DH44$^{PI}$N ISIs differed significantly between starved and fed flies: ISIs were shifted towards longer durations after starvation (p<<0.001, *Figure 3—figure supplement 1B*). Hence, while the overall spike frequency was unaffected, and the baseline membrane potential remained unchanged (*Figure 3—figure supplement 1C*), DH44$^{PI}$N spike patterns were more irregular and burst-like after feeding.

Finally, we tested whether DH44$^{PI}$Ns are responsive to glucose perfusion. Following the exact same protocol we used for IPC recordings, we perfused the brain with saline containing 40 mM glucose, while patching from DH44$^{PI}$Ns. The spike rate of DH44$^{PI}$Ns significantly increased during perfusion with 40 mM glucose in starved flies, by about 0.5 Hz (*Figure 3G*). The ISI distribution was also significantly shifted towards shorter intervals during glucose perfusion across all flies and within each recording (*Figure 3—figure supplement 1D and E*). Fed flies, on the other hand, did not exhibit a significant change in DH44$^{PI}$N activity during glucose perfusion (*Figure 3—figure supplement 1F*). This indicated that DH44$^{PI}$Ns are indeed sensitive to glucose perfusion, and that the glucose sensitivity depends on the nutritional state, as previously reported (*Dus et al., 2015*). Together, these experiments reveal two important points: First, DH44$^{PI}$Ns are glucose sensing, whereas IPCs are not (*Figure 3A*). Second, the DH44$^{PI}$N recordings confirm that our negative result for IPC glucose sensitivity was not an experimental artifact.

## DH44Ns outside the PI inhibit IPCs

The proximity of DH44$^{PI}$N and IPC somata in the same cluster of the PI, their overlapping neurites in the brain (*Figure 3D*), and the fact that both populations are implicated in metabolic homeostasis prompted us to determine whether these neurons directly interact with each other. To investigate potential functional connectivity between DH44Ns and IPCs, we combined the *UAS-GAL4*

and *LexA-LexAop* systems to simultaneously express CsChrimson in all DH44Ns and GFP in IPCs (*Figure 4A*, see also *Figure 4—figure supplement 1A*; *Liessem et al., 2023*). This approach enabled us to optogenetically activate DH44Ns while recording from IPCs via patch-clamp. Optogenetic activation of DH44Ns drove an immediate, strong hyperpolarization of IPCs, which lasted for >1 s after stimulus cessation and reduced IPC spiking repeatedly across trials (*Figure 4B*). This effect was robust across the population of IPCs and flies (*Figure 4C*). On average, the IPC membrane potential dropped by –5 mV below the baseline during DH44N activation and stayed hyperpolarized until about 1 s after activation (*Figure 4C*). This fast and strong response suggests that DH44Ns could be directly presynaptic to IPCs. To quantify this effect, we calculated the median $V_m$ for each IPC recording in a 500ms window before the stimulus onset (pre, gray box in *Figure 4C*) and 200 ms after cessation of the stimulus (post, purple box in *Figure 4C*). All IPCs recorded were strongly hyperpolarized by DH44N activation (*Figure 4D*, p=0.002). Hence, DH44Ns strongly and significantly inhibited IPCs. Notably, we also recorded from one IPC, which was excited during DH44N activation and inhibited immediately after cessation of activation (*Figure 4—figure supplement 1C*), indicating heterogeneity in the IPC population (see also *Held et al., 2024*). Overall, our results indicate that acute activation of DH44Ns inhibits IPC activity, hence regulating insulin release. This strong inhibition is surprising, since our previous results implicated both, the DH44$^{PI}$Ns and IPCs, in glucose sensing and metabolic control. Hence, our assumption had been that these modulatory neurons would display reciprocal excitation rather than inhibition. However, the driver line we used for DH44N activation was relatively broad, and any of the neurons labeled could be responsible for the IPC inhibition (*Figure 4A*). For example, the line labels a subset of DH44Ns ascending from the VNC. A pair of these DH44Ns co-express the neuropeptide Leucokinin (LK) (*Zandawala et al., 2018*), and LK-expressing neurons inhibit most IPCs (*Held et al., 2024*).

To investigate this, we identified a driver line which only labels the DH44Ns in the PI (*Asahina and Anderson, 2013*) (DH44$^{PI}$–GAL4) and repeated the functional connectivity experiments using this more specific driver line (*Figure 4E*, see also *Figure 4—figure supplement 1B*). Activating exclusively DH44$^{PI}$Ns had no effect on the IPC membrane potential or spike rate (*Figure 4F, and G*: grand mean remained 0 mV). This was consistent across the population of IPCs and flies (*Figure 4H*, p=0.641). These results confirm that the strong inhibition of IPCs observed when activating the broad DH44N line is not driven by DH44$^{PI}$Ns. Therefore, DH44$^{PI}$Ns and IPCs both sense changes in the nutritional state - IPCs after feeding and DH44$^{PI}$Ns via hemolymph sugar concentrations - but act independently and in parallel to signal these changes (*Figure 3H*).

## Activation of DH44Ns modulates locomotor activity

Given their sensitivity to glucose and connectivity to IPCs, DH44Ns were strong candidates to contribute to the effects of nutritional state changes on locomotor activity (*Figure 2*). DH44Ns were previously shown to promote feeding in starved animals (*Dus et al., 2015*). In particular, we hypothesized that the broad DH44N line could have opposite effects on IPCs since a subset of them inhibited IPCs. We, therefore, tested whether DH44Ns affect locomotor activity by following the same approach as for IPC and OAN activation in freely walking flies (*Figure 2C–K*). Activation of the broad DH44N line resulted in a dramatic increase in FV from a median of 1.8 mm/s to 6 mm/s immediately after stimulus onset (*Figure 4I and J*). This increase was followed by an abrupt halt that lasted until DH44N activation ceased (*Figure 4I–K*). During this time window of reduced locomotor activity, we observed a strong increase in proboscis extension (PE) in 6/20 flies for which we quantified this (*Figure 4—figure supplement 1D*). The locomotor arrest was released soon after cessation of DH44N activation, upon which locomotor activity increased until the onset of the next stimulation period (*Figure 4I and K*). The FV in the post-activation window increased over subsequent activation cycles and was significantly higher than in controls throughout the experiment (*Figure 4I and K*). The FV of DH44N-activated flies increased from a median of 0.3 mm/s to 3 mm/s over five activation cycles, suggesting that long-term activation of DH44Ns leads to an increase in locomotion comparable to starvation-induced hyperactivity.

In addition, we tested the driver line labeling only DH44$^{PI}$Ns. Although we did observe an overall increase in the FV of the experimental group, we neither observed an activity peak upon activation onset, nor locomotor arrest during activation (*Figure 4L–N*). Furthermore, contrary to the activation of the broad DH44 driver line, specific activation of the DH44$^{PI}$Ns did not drive PE (*Figure 4—figure*

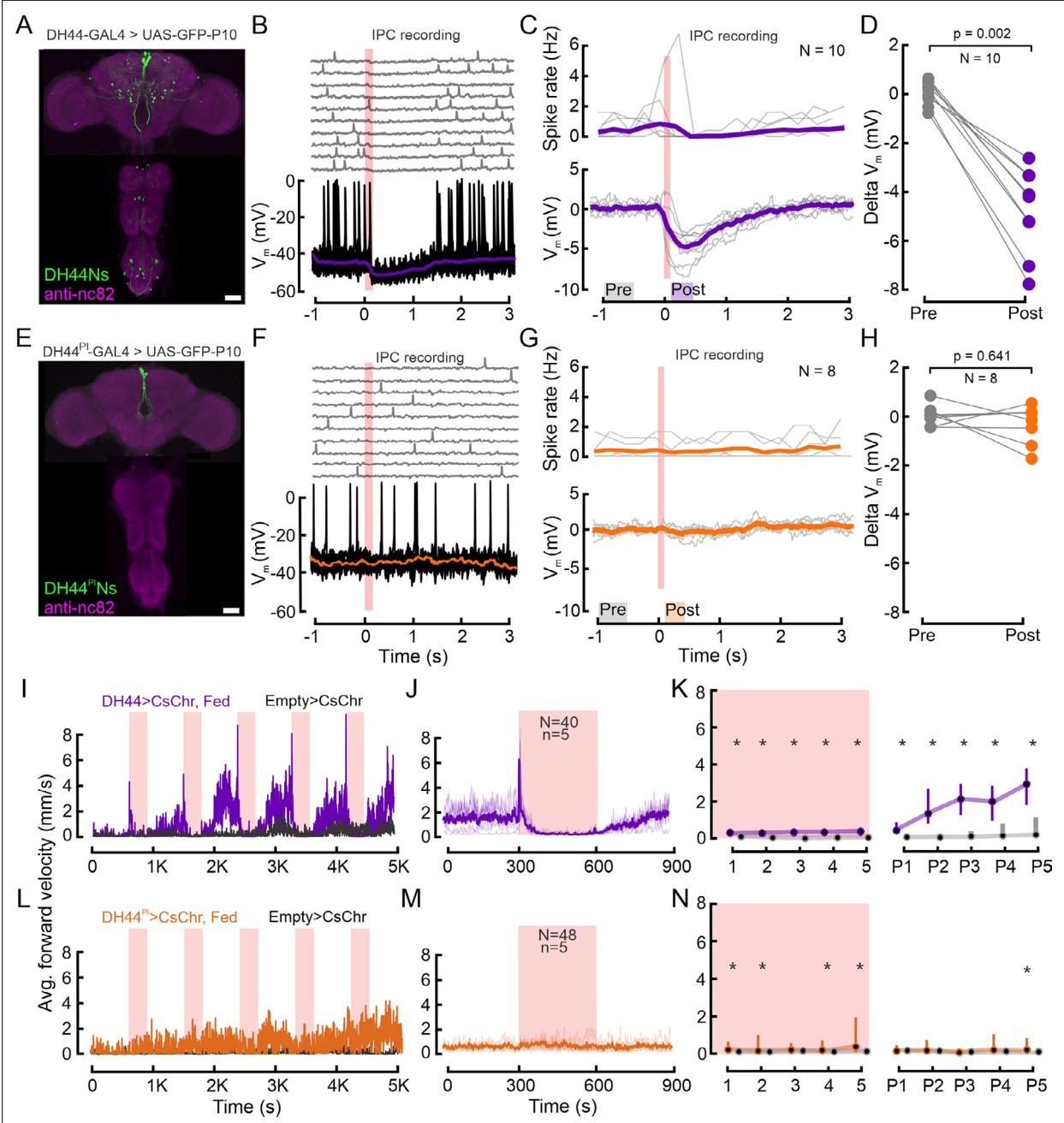

**Figure 4.** DH44Ns outside the pars intercerebralis (PI) inhibit insulin-producing cells (IPCs) and drive increases in locomotor activity. (**A**) Immunolabeling showing DH44 expression in the brain and the VNC in the broad *DH44-GAL4* driver line. GFP was enhanced with anti-GFP (green), and brain and VNC neuropils were stained with anti-nc82 (magenta). (**B**) Example recording of an IPC during optogenetic activation of the DH44Ns (red shading). Upper panel shows individual trials, lower panel shows ten trials overlaid, and the median of all trials (purple trace). (**C**) Upper panel shows the spike density of individual IPCs across 10 DH44N activations. Lower panel shows the baseline-subtracted, median-filtered $V_m$ traces for each IPC. Thick lines represent the grand mean. (**D**) Effect of DH44N activation on IPCs. Delta $V_m$ is plotted by calculating the median baseline subtracted $V_m$ from C) 500 ms before (Pre) and 200 ms after DH44N activation (Post). Each circle represents one IPC recording. p-values from the Wilcoxon signed-rank test. (**E**) Immunolabeling showing GFP expression in the brain and the VNC in the sparse *DH44^{PI}-GAL4* driver line. (**F**) Example recording of an IPC during optogenetic activation of DH44 neurons using the sparse DH44^{PI}N driver line. Plot details as in B. (**G**) Spike density and baseline subtracted median $V_m$ of individual IPCs during activation of the DH44^{PI}N driver line. Plot details as in C. (**H**) Pre and post-delta $V_m$ of IPCs before and after optogenetic activation of the DH44^{PI}N line. Plot details as in D. See *Supplementary file 1e* for a number of flies. (**I**) Average forward velocity (FV) of 20 flies during optogenetic activation of the DH44Ns using the broad driver line (see A). *Empty split-GAL4* was used as a control for all experiments (black). Red shading indicates 300 s optogenetic activation windows. (**J**) Average FV across all DH44N activation trials based on two independent replications of the experiment in I. Note that the peak in average FV lies within the first frame of the stimulation window. (**K**) Average FV of all flies during each stimulus trial

*Figure 4 continued on next page*

*Figure 4 continued*

(1-5) and post-stimulus trial (300 s window immediately after activation ceased, (P1–P5). Circles and bars show median and IQR, respectively. Asterisks represent a significant difference according to a Wilcoxon rank-sum test. (**L–N**) Behavioral effects of optogenetic DH44^PI^N activation (see E). Plot details as in I, J, and K, respectively. See *Supplementary file 1c and d* for p-values. Scale bars: 50 μm.

The online version of this article includes the following figure supplement(s) for figure 4:

**Figure supplement 1.** Effects of differential activation of DH44Ns and DH44^PI^Ns on insulin-producing cells (IPCs) and behavior.

*supplement 1E*). Hence, the strong activation phenotypes were driven by DH44Ns outside the PI. We conclude that the activation of DH44Ns, especially the broad DH44N line, had a pronounced and sustained effect on foraging behavior. The same neurons strongly and significantly inhibited IPCs, suggesting that feeding-promoting and satiety-inducing pathways form inhibitory connections in the CNS as a part of the neuronal mechanisms to ensure reciprocal inhibition of satiety and hunger states.

## Discussion

### IPC activity is modulated on short and long timescales by the nutritional state, the behavioral state, and aging processes

To advance our understanding of the neuronal underpinnings of metabolic homeostasis, we analyzed the nutritional state-dependent modulation of IPCs in *Drosophila*. Our in vivo patch clamp recordings revealed that IPC activity is diminished during starvation and rebounds after glucose feeding. This is consistent with previous studies demonstrating an accumulation of DILPs (*Géminard et al., 2009*; *Ikeya et al., 2002*; *Rajan and Perrimon, 2012*; *Bai et al., 2012*; *Sudhakar et al., 2020*; *Claessens et al., 2008*) and a reduction of *dilp* transcript levels (*Rajan and Perrimon, 2012*; *Sudhakar et al., 2020*; *Claessens et al., 2008*) in IPCs of adult and larval fly brains under starvation and underlines that electrophysiological IPC activity and DILP release are correlated. However, two aspects were surprising about our results. First, the rebound after glucose feeding was relatively slow, and IPC activity built up to baseline levels after over 6 h of refeeding. These slow temporal dynamics are in stark contrast to the behavioral state-dependent modulation of IPC activity, which occurs on a timescale of milliseconds (*Liessem et al., 2023*). This suggests that IPCs integrate changes in internal states over different timescales and potentially affect metabolism and neuronal networks on both short and long timescales. The second surprising result was that the reduced IPC activity after starvation failed to rebound upon refeeding with high protein and standard diets (*Figure 1H*). Our standard diet contains glucose at very low concentrations (upper bound of 30 mM, see Methods for details), which partially explains the lack of an IPC activity rebound. However, the observation that IPC activity in flies refed with SD remained comparable to that of starved flies suggests that starvation had a long-lasting effect on IPC activity, which could specifically be restored by feeding nutritive sugars (glucose and fructose). One potential explanation could be that older flies exhibited lower IPC activity levels, which could lower the baseline activity after refeeding and thus counteract the rebound in IPC activity. However, we always recorded from flies aged between 5–6 d in refeeding experiments, and a significant decrease in IPC activity was only observed in flies older than 8 d (see *Figure 1—figure supplement 2B*). This suggests that aging played only a minor part here. Instead, reduced IPC activity after refeeding with SD may indicate increased insulin sensitivity as a result of the starvation-refeeding paradigm. Increased insulin sensitivity has been reported in multiple mammalian studies which used a starvation-refeeding paradigm to study effects of dietary restriction on insulin release (*Sutton et al., 2018*; *Lu et al., 2011*), and our results indicate that this could be similar in flies. DILP6 mRNA in the fat body is upregulated during starvation, which has been shown to decrease systemic insulin signaling by the suppression of insulin release from IPCs (*Bai et al., 2012*), potentially contributing to mechanisms regulating insulin sensitivity in flies. Furthermore, we observed that IPC activity did not recover on the HP diet, which also resulted in poor survival rates. This may indicate that insulin signaling is impaired in flies if they only feed on protein. Similar observations were made in mammals, where prolonged high-protein diets lead to insulin resistance and type II diabetes (*Claessens et al., 2008*; *Rietman et al., 2014*).

One of our experiments demonstrated that IPC activity was heavily diminished in flies older than 10 d (*Figure 1—figure supplement 2B*). A possible explanation could be that flies feed less as they

age. However, this only holds true for flies older than 14 d (*Wong et al., 2009*). Therefore, reduced IPC activity in 10–11 day-old flies is unlikely to result from reduced food intake and likely involves inhibition of insulin signaling. For instance, *dilp2* mutants show an increased lifespan (*Grönke et al., 2010*) which could partially result from elevated glycogen phosphorylase activity (*Post et al., 2018*), an enzyme involved in regulating glycogen storage. This suggests that suppressed IPC activity in aged flies could mediate glucose homeostasis by increasing glycogen catabolism. Reduced insulin signaling has previously been linked to increased life-span in different species, including yeast, worms, flies, and rodents (*Barbieri et al., 2003*). The suppression of IPC activity could be a mechanism to increase survival under physiologically unfavourable conditions. In summary, the reduced IPC activity we observed in starved and aged flies may be crucial for metabolic adjustments required for survival. Notably, similar mechanisms have been observed in mammals, where reduced insulin signaling and increased insulin sensitivity during fasting or dietary restriction contribute to enhanced metabolic efficiency and longevity. These parallels suggest that the regulation of insulin signaling in response to nutrient availability is an evolutionarily conserved strategy, underscoring its fundamental role in survival and metabolic homeostasis across species.

### The nutritional state-dependent modulation of IPCs supports the presence of an incretin-like effect in *Drosophila*

Mammalian pancreatic beta cells sense glucose levels in the blood and release insulin accordingly (*Matschinsky, 1996*; *MacDonald et al., 2005*). However, this process is much more complex than initially assumed. One key observation is that the ingestion of glucose drives a much higher release of insulin compared to the isoglycemic intravenous perfusion of glucose. This difference in insulin release is partially driven by incretin hormones from the gut, for example, glucagon-like peptide-1 (GLP-1) and glucose-dependent insulinotropic peptide (GIP) (*Buchan et al., 1978*; *Meier et al., 2002*; *Brown, 1982*), which are released after the ingestion of food. Hence, insulin release is modulated by enteroendocrine signals. *Drosophila* IPCs are functionally analogous to mammalian beta cells and presumably sense glucose in the circulating hemolymph to release insulin in response to increases in glucose concentration. Previous ex vivo studies suggested that IPCs, like pancreatic beta cells, sense glucose cell-autonomously (*Fridell et al., 2009*; *Kréneisz et al., 2010*). Consistent with this, we observed an increase in IPC activity after the ingestion of glucose (*Figure 2B*). However, IPC activity did not increase during the perfusion of glucose directly over the brain. Importantly, the fly preparations were kept alive for several hours allowing the glucose-rich saline to enter circulation and reach all body parts. Several factors may explain the difference between ex vivo and in vivo preparations. First, in ex vivo studies, certain regulatory feedback mechanisms present in vivo could be absent. For example, the strong inhibitory input IPCs received from DH44Ns we found would likely be absent in brain explants without a VNC. A lack of inhibitory feedback might allow for more direct glucose sensing by IPCs ex vivo, whereas in vivo, the IPC response could be suppressed by more complex systemic feedback. Second, we attempted to use the intracellular saline formulation employed in a previous ex vivo study (*Kréneisz et al., 2010*). However, we observed that IPCs depolarized quickly using this saline, leading to unstable recordings that did not meet our quality standards for in vivo experiments. Another possible explanation for the lack of an effect of glucose might have been that the dominant circulating sugar in flies is trehalose (*Wyatt and Kalf, 1957*; *Elbein et al., 2003*) which is derived from glucose. When we extended our experiments, we found that trehalose perfusion did not affect IPC activity either, strengthening the idea that IPCs do not directly sense changes in hemolymph sugar levels. Therefore, our findings suggest that, similar to mammals, IPC activity and hence, insulin release, is not simply modulated by hemolymph sugar concentration in *Drosophila*. Several pathways could contribute to this. First, although an incretin effect has not been reported in flies before, recent studies provided evidence of incretin-like hormones such as the midgut-derived neuropeptide F in flies (*Yoshinari et al., 2021*; *Malita et al., 2022*), suggesting that a similar signaling pathway could underlie the incretin effect across species. Second, a subset of IPCs was shown to express the mechanosensitive channel piezo and innervate the crop (*Wang et al., 2020*), suggesting that an increase in crop volume during feeding could contribute to the increase in IPC activity after feeding. However, the fact that we observed strong responses of IPCs to glucose intake but not the ingestion of regular food during refeeding suggests this is not the core mechanism underlying the incretin-like effect we observed. Third, taste pathways converge onto IPCs (*Yao and Scott, 2022*) and they could increase

IPC activity after the ingestion of glucose via serotonergic inputs from taste-responsive cells in the gnathal ganglion. Here, however, we would expect faster responses on the scale of seconds, which are unlikely to shift IPC activity over several hours. Hence, gut-secreted neuropeptides are the strongest candidates to mediate the incretin effect in *Drosophila*. These results signify a causal link between glucose homeostasis in the gut and the brain and pave the way for further studies investigating the evolutionary conservation of the incretin effect across mammals and insects.

We observed that IPC activity increased over a timescale of hours, which is longer compared to the fast insulin response in mammals, where insulin typically peaks within an hour of feeding (*Holst, 2019*). In flies, insulin levels rise within minutes of refeeding, followed by a drop after 30 min (*Park et al., 2014*). Our experimental techniques limit our ability to capture these fast initial dynamics, since the preparation for intracellular recordings requires tens of minutes, so that we typically recorded IPC activity at least 20 min after the last food ingestion. Notably, studies in fasted mammals have shown that insulin peaks within minutes of refeeding, followed by a rapid decline, with levels stabilizing as feeding continues (*Boland et al., 2018*; *Kubota et al., 2008*). We speculate a similar dynamic could be present in flies, but with our approach, we capture the steady-state reached tens of minutes after food ingestion rather than a potential initial peak.

## DH44$^{PI}$Ns and IPCs have distinct, yet interrelated roles in nutrient sensing

Our in vivo recordings revealed that IPCs do not sense glucose cell-autonomously, and neither do neurons in the CNS that are directly connected to IPCs – otherwise, we would have expected an increase in IPC activity during glucose perfusion. In contrast, DH44$^{PI}$Ns were sensitive to changes in the extracellular glucose concentration. This could either be driven by cell-autonomous glucose sensitivity or presynaptic glucose-sensing neurons. Moreover, DH44$^{PI}$Ns did not show an increase in activity in fed flies, indicating a lack of sensitivity to glucose ingestion. IPCs responded to glucose ingestion on a timescale of hours, whereas DH44$^{PI}$Ns immediately responded to increases in extracellular glucose concentrations on a timescale of seconds to minutes. These results suggest that DH44$^{PI}$Ns and IPCs play distinct roles in glucose sensing, which might complement each other. Our initial functional connectivity experiments suggested that the glucose-sensing DH44$^{PI}$Ns might inhibit IPCs directly. However, refined experiments using a sparse driver line revealed that IPCs are inhibited by DH44Ns outside the PI. If these DH44Ns, like DH44$^{PI}$Ns, are glucose sensing they would suppress IPC activity in the presence of high glucose concentrations, which would be counteractive. Therefore, even though this remains to be determined, it is likely that these non-PI DH44Ns are activated by signals other than glucose concentrations. Notably, the DH44$^{PI}$Ns express the DH44 peptide, as confirmed by anti-DH44 stainings (*Lee et al., 2015*). This also applies to a large fraction of neurons labeled in the broad DH44 driver line (*Lee et al., 2015*). However, a subset of neurons labeled in the broad line did not exhibit DH44 immunoreactivity (*Lee et al., 2015*), and might, therefore, not actually express the DH44 peptide. Hence, the inhibition of IPCs could be driven by neurons in the DH44 driver line that do not express DH44. A strong candidate for the inhibition is LK and DH44-positive neurons, which are labeled by the broad line (*Zandawala et al., 2018*). In a parallel study, we showed that LK-expressing neurons strongly inhibit IPCs (*Held et al., 2024*), similar to the broad DH44 line used here. Furthermore, evidence from single-nucleus transcriptomic analysis shows that IPCs express DH44-R1 and DH44-R2 receptors (*Held et al., 2024*). Therefore, it is possible that DH44Ns communicate with IPCs through a direct peptidergic connection. Notably, the inhibitory effect of non-PI DH44Ns on IPCs was very strong and fast, suggesting that a connection via classical synapses is more likely. Regardless, our results show that the glucose-sensing DH44$^{PI}$Ns and IPCs act independently of each other.

In line with previous studies (*Dus et al., 2015*; *Oh et al., 2021*), we found that DH44$^{PI}$Ns were only activated by glucose in starved flies. Interestingly, this was not because the DH44$^{PI}$N spike rate was lower after starvation. Instead, DH44$^{PI}$Ns exhibited similar spike rates in fed and starved flies, but the activity increase in response to glucose perfusion only occurred under starvation. DH44$^{PI}$Ns also displayed a level of modulation beyond simple increases and decreases in spike frequency. In particular, the ISI distributions of DH44$^{PI}$Ns were shifted to significantly longer intervals during starvation. This may contribute to the increased glucose sensitivity in starved flies. Moreover, shorter ISIs in fed flies and the more bursty activity pattern could lead to an increase in DH44 peptide release despite similar overall spike frequencies. Apart from this, DH44$^{PI}$Ns themselves could also become

more sensitive to glucose in starved animals. This would suggest that DH44$^{PI}$Ns sense changes in glucose concentration, rather than the absolute concentration of glucose, in the brain or hemolymph.

## Nutritional state-dependent modulation of behavior

Changes in the activity of modulatory neurons affect information processing in neuronal circuits and ultimately behavior (*Nässel and Winther, 2010*; *Ko et al., 2015*; *Marder, 2012*; *van den Pol, 2012*; *Asahina et al., 2014*). This includes starvation-induced hyperactivity and other aspects of foraging in *Drosophila*. In this regard, we first tested whether IPCs modulate locomotor activity in flies. This was prompted by the observation that starvation strongly modulated both IPC activity and locomotion. However, IPCs only had a minor effect on locomotion. This was despite stimulating IPCs using long, high-intensity optogenetic activation. The strongest effect we observed was a reduction in starvation-induced hyperactivity upon IPC activation, which we interpret as a satiety-like effect due to increased IPC activity. This effect was stronger at the beginning of the experiment, even before the first activation of IPCs, and was robust across independent repetitions of the same experiment. This was likely in part due to an increased IPC depolarization via CsChrimson activation because of exposure to white light during the preparation, recovery period, and behavioral experiment. We carried out all behavioral experiments in low-intensity white light. Background light levels strongly affect locomotor activity in flies (*Helfrich-Förster, 2001*), and overall we observed more reliable starvation-induced hyperactivity under light conditions. We suspect the optogenetic effect of background light was small, though, because in genotypes that had very strong and acute effects on locomotion under red light the baseline locomotor activity under background light conditions was in the range of controls (*Figures 2I–K, and 4I–K*). Another reason why we performed experiments under light conditions and not in the dark is that switching from dark to light leads to a startle response in optogenetic activation experiments when the red LED is switched on, which masks behavioural responses induced by neuronal activation. We minimized this effect by using low intensity white background light. With subsequent activations, the difference between IPC-activated and control flies was reduced, probably because flies were increasingly food and water-deprived after an hour of activation experiments. In particular the latter could potentially override the reduction in starvation-induced hyperactivity induced by optogenetic IPC activation. Overall, our experiments revealed that IPC activation and resulting insulin release do not lead to a sustained satiety response observable in behavior. This aligns with physiological expectations, as elevated insulin levels exacerbate the glucose deficit in starved flies, which could ultimately lead to increases in hyperactivity via this secondary effect.

In contrast to insulin, we confirmed the causal role of one of the known neuromodulators involved in driving starvation-induced hyperactivity – octopamine. As expected, OAN activation increased locomotor activity substantially, which we interpret as a proxy for foraging based on previous publications (*Yang et al., 2015*; *Huang et al., 2020*). The activity levels reached by long-term OAN activation were comparable to the effects of long-term starvation. These results underscore that octopamine plays a major role in driving starvation-induced hyperactivity. Our results also suggest a key role of DH44Ns in modulating aspects of foraging behavior. There is evidence that DH44 release in the brain modulates rest-activity rhythms (*Cavanaugh et al., 2014*) via downstream targets in the gnathal ganglion (*King et al., 2017*), which houses circuits involved in the modulation of feeding and locomotion (*Marella et al., 2012*; *King et al., 2017*; *Singh, 1997*; *Sterne et al., 2021*; *Colomb et al., 2007*; *Tastekin et al., 2015*; *Cobb et al., 2009*). Our optogenetic approach provides substantial evidence that DH44Ns drive increased locomotor activity in adult flies, plausibly to enable exploration of the environment for locating food sources. First, the increase in locomotion phenocopies starvation-induced hyperactivity. Second, our experiments additionally showed that DH44Ns outside the PI drive proboscis extensions during stopping, which are a hallmark of feeding behavior. In line with this, a previous study has shown that activation of DH44 receptor 1 neurons, the target neurons of the DH44 neuropeptide, induces proboscis-extension (*Dus et al., 2015*). Both observations suggest that DH44Ns are a component of feeding pathways and mediate nutrient uptake in order to maintain nutritional homeostasis.

Our functional connectivity experiments revealed that these DH44Ns inhibit IPC activity. At first glance, promoting feeding and inhibiting insulin release at the same time seems counter-regulatory. However, there are two possible explanations: First, these two effects could be driven by different sub-populations of DH44Ns labeled by the line we used (*Figure 4A and E*). Second, the IPC inhibition could be part of a combination of physiological effects signaling a hunger state, and thus contribute

to the increased foraging activity driven by DH44Ns. Notably, blocking insulin signaling has been reported to mimic starvation in fed flies (*Ko et al., 2015*). Contrary to DH44Ns outside the PI, activation of the glucose-sensing DH44$^{PI}$Ns did not drive proboscis extension, suggesting that DH44$^{PI}$Ns are not involved in feeding initiation, but primarily function in post-feeding regulation of metabolic homeostasis. This is in line with our electrophysiological results showing that DH44$^{PI}$N spike frequency is not affected by starvation, and only starts to increase once glucose levels change in the hemolymph. Furthermore, DH44$^{PI}$N activation slightly increased the FV but did not lead to stopping, suggesting that DH44$^{PI}$Ns contribute to starvation-induced hyperactivity, whereas other DH44Ns drive stopping and proboscis extension.

In conclusion, our study sheds light onto the intricate connections between neuroendocrine signaling, nutrient sensing, and behavior in *Drosophila*. Using electrophysiological approaches, we unraveled the complex activity dynamics of IPCs and DH44$^{PI}$Ns and identified differences in their activity patterns during various nutritional states. The discovery of an 'incretin-like' effect in flies suggests that important aspects of gut-brain signaling are conserved across vertebrate and invertebrate species. Hence, our findings not only contribute to the increasing body of research on insulin signaling, but also pave the way for future research analyzing the intricacies of *Drosophila* neuroendocrine networks governing nutrient sensing and behavior.

# Materials and methods

## Key resources table

| Reagent type (species) or resource | Designation | Source or reference | Identifiers | Additional information |
|---|---|---|---|---|
| Strain (*D. melanogaster*) | DILP2-GAL4; CyO | Bloomington *Drosophila* Stock Center | BDSC:37516 | Used to drive expression in IPCs |
| Strain (*D. melanogaster*) | 10XUAS-IVS-myr::GFP | Bloomington *Drosophila* Stock Center | BDSC:32197 | Used to express GFP |
| Strain (*D. melanogaster*) | TDC2-GAL4 | Bloomington *Drosophila* Stock Center | BDSC:9313 | |
| Strain (*D. melanogaster*) | DH44-GAL4 | Bloomington *Drosophila* Stock Center | BDSC:39347 | |
| Strain (*D. melanogaster*) | DH44PI-GAL4 | Bloomington *Drosophila* Stock Center | BDSC:51987 | |
| Strain (*D. melanogaster*) | R96A08-LexA-p65-vk37::LexOp-dilp2::GFP; 20x-UAS-CsChrimson-attp2/TM6b | *Liessem et al., 2023* | As described in the publication | Used for optogenetics experiments to drive CsChrimson in Gal4 driver lines while simultaneously expressing GFP in IPCs |
| Strain (*D. melanogaster*) | 10X-UAS-GFP-P10 | Bloomington *Drosophila* Stock Center | BDSC:32201 | |
| Strain (*D. melanogaster*) | 20X-UAS-CsChrimson | Bloomington *Drosophila* Stock Center | BDSC:55134 | |
| Strain (*D. melanogaster*) | Empty split-GAL4 | Bloomington *Drosophila* Stock Center | BDSC:86738 | |
| Chemical compound | all-trans-retinal | Sigma-Aldrich | R2500 | Used in optogenetic experiments |
| Software, algorithm | pCLAMP 10 | Molecular Devices | RRID:SCR_011323 | Used for electrophysiological recordings |
| Software, algorithm | MATLAB R2021a | The Mathworks | RRID:SCR_001622 | Used for data analysis and statistical testing |
| Software, algorithm | OCULAR | OCULAR | RRID:SCR_024467 | Image acquisition software |
| Antibody | anti-DILP2 (primary Rabbit polyclonal) | A. Veenstra, Bordeaux, FR | RRID:AB_2569969 | Used to label IPCs, diluted 1:2000 |
| Antibody | anti-nc82 (primary Mouse monoclonal) | DSHB | RRID:AB_2314866 | Used to label neuropils, diluted 1:500 |
| Antibody | anti-GFP (primary Chicken polyclonal) | Abcam | ab13970, RRID:AB_300798 | Enhances GFP signal, diluted 1:1000 |
| Chemical compound | Vectashield Antifade Mounting Medium | VEC-H-1000 | Biozol | |
| Antibody | Goat anti-chicken Alexa Fluor 488 (secondary goat polyclonal) | Thermo Fisher Scientific | RRID:AB_2534096 | 1:200 |
| Antibody | Goat anti-rabbit Alexa Fluor 555 (secondary goat polyclonal) | Thermo Fisher Scientific | RRID:AB_2535850 | 1:200 |
| Antibody | Goat anti-mouse Alexa Fluor 635 (secondary goat polyclonal) | Thermo Fisher Scientific | RRID:AB_2536184 | 1:400 |
| Chemical compound | SigmaCote | Sigma-Aldrich | cat. no. SL2; RRID:SCR_008988 | Siliconizing reagent |
| Software, algorithm | Fiji | *Schindelin et al., 2012* | RRID:SCR_002285 | Used for image processing |

## Fly husbandry

The following fly strains were used: *DILP2-GAL4; CyO* (BDSC #37516), *10XUAS-IVS-myr::GFP* (BDSC #32197), *TDC2-GAL4* (#BDSC 9313), *DH44-GAL4* (BDSC #39347), *DH44^{PI}-GAL4* (BDSC #51987), *R96A08-LexA-p65-vk37::LexOp-dilp2::GFP; 20x-UAS-CsChrimson-attp2/TM6b* (as described in *Liessem et al., 2023*), 10X*UAS-GFP-P10* (BDSC #32201), *20x-UAS-CsChrimson* (BDSC #55134), *Empty split-GAL4* (BDSC #86738). *DILP2-GAL4* and *DH44-GAL4* driver lines were crossed with *10XUAS-IVS-myr::GFP* to drive the expression of GFP in IPCs and DH44Ns, respectively, for patch clamp recordings. For electrophysiology combined with optogenetics, DH44N driver lines were crossed with *R96A08-LexA-p65-vk37::LexOp-dilp2::GFP; 20x-UAS-CsChrimson-attp2/TM6b*, hence expressing CsChrimson in DH44Ns and GFP in IPCs.

Flies were raised at 25 °C and 60% humidity under a 12 h/12 h light/dark cycle, on standard fly food containing per liter: 147.5 g cornmeal, 10 g soy flour, 18.5 g Cenovis (beer yeast), 6.25 g agar-agar, 45 g malt syrup, 45 g sugar beet molasses, and 2.5 g Nipagin. For starvation, flies were kept in empty vials with a filter paper soaked in water. For refeeding experiments, starved flies were refed on one of the following diets: 400 mM D-glucose in 2% agar (HG), 400 mM D-glucose in standard diet (HG + SD), 400 mM D-fructose in 2% agar (HF), 400 mM D-arabinose in standard diet (HA + SD), 10% yeast extract in 2% agar (HP) or standard fly food (standard diet). For optogenetic experiments, 300 µM of all-trans-retinal (R2500, Sigma-Aldrich, Steinheim, DE) was added to 10 ml of fly food. These vials were kept in darkness until the flies were used for experiments. Our standard lab diet does not contain glucose as such, but we estimated the concentration of glucose based on indirect sources of glucose (malt syrup and sugar beet molasses). Based on the percentage of glucose present in malt syrup (7%) and sugar beet molasses (4.55 g per 100 g), we estimate that our lab diet contains about 28 mM of glucose.

## Electrophysiology

All experiments were performed in mated females between 3–6 d post eclosion, except for the comparison of IPC activity between different age groups and mating states. Flies were cold anesthetized on ice and then immobilized in a custom-made shim plate fly holder using UV glue (Proformic C1001, VIKO UG, Munich, DE). The proboscis was glued to the thorax to restrict brain movement, and the front legs were excised. The cuticle was then removed in a window above the pars intercerebralis so that the posterior-dorsal part of the fly brain was exposed. Trachea and the ocellar ganglion were removed to render the IPCs and DH44^{PI}Ns accessible. The fly was then transferred with the fly holder into a customized, upright fluorescence microscope setup (Olympus BX51WIF, Evident Corporation, Tokyo, JPN). Live images of the fly brain were acquired with a high-resolution camera (SciCam Pro, Scientifica, Uckfield, UK) using an image acquisition software (OCULAR, Digital Optics Limited, Auckland, NZ).

During the preparation, as well as the experiment, the brain was continuously perfused with carbonated (95% $O_2$ and 5% $CO_2$) extracellular saline containing, 103 mM NaCl, 3 mM KCl, 5 mM N-[Tris(hydroxymethyl)methyl]–2-aminoethanesulfonic acid, 20 mM sucrose, 26 mM NaHCO3, 1 mM NaH2PO4, 1.5 mM CaCl2.2H2O, 4 mM MgCl2.6H2O, and osmolarity adjusted to 273–275 mOsm (modified from *Gouwens and Wilson, 2009*). 0.025% Collagenase (w/v in extracellular saline, Sigma-Aldrich #C5138) was gently applied using a thin-walled glass pipette to dissociate the neural sheath above cell bodies before recordings. IPC/DH44^{PI}N cell bodies were identified via GFP expression and whole-cell patch clamp recordings were performed using thick-walled patch pipettes (4–8 MΩ resistance) containing intracellular saline (40 mM potassium aspartate, 10 mM HEPES, 1 mM EGTA, 4 mM MgATP, 0.5 mM Na3 GTP, 1 mM KCl and 20 µM, 265 mOsm, pH 7.3). Continuous time series recordings of the membrane potential were captured in current clamp mode with an AxoPatch Multi-Clamp 200B (Molecular Devices, Sunnydale, CA, USA) and corrected for a 13 mV liquid junction potential (*Gouwens and Wilson, 2009*). All data were recorded with a Digidata 1440 A analog-digital converter (Molecular Devices), controlled by the pCLAMP 10 software using a 10 kHz low-pass filter and a 20 kHz sampling rate. Recordings were accepted for analysis if the resting membrane potential was <–48 mV and the spike amplitude >20 mV.

Baseline activity was always recorded in glucose-free saline for 10 min, and the last 5 min were used for analysis. For glucose perfusion experiments, IPC activity was recorded in glucose-free extracellular saline for 10 min to establish the baseline. Next, the brain was perfused with extracellular saline

replacing 40 mM sucrose with 40 mM glucose. Similarly, trehalose-rich saline was prepared by substituting 40 mM sucrose with 20 mM trehalose to maintain consistent osmolarity. The switch between the salines took about three min, at which point the glucose/trehalose-rich saline reached the brain. The membrane potential was then recorded for 15 min in three consecutive 5-min recordings. Out of these, the last two recordings (Glu1/Tre1 and Glu2/Tre2) were used for analysis to make sure that glucose had time to reach the IPCs and deeper structures inside the brain.

For optogenetic activation experiments during electrophysiological recordings, CsChrimson was activated using a 625 nm LED with an intensity adjusted to ~4.4 mW/cm$^2$ at the position of the fly. The activation protocol was triggered by the pCLAMP software. Each recording consisted of 10 activations with the LED ON for 100 ms, interleaved with a 10 s long inter-stimulus-interval. TTL triggers were used to drive the LED and were recorded simultaneously with intracellular traces in the abf format (Axon Binary File).

## Quantification and statistical analysis for electrophysiology

All data analysis was performed in MATLAB R2021a (The Mathworks, Natick, MA, USA). The statistical tests we used, and respective p-values are mentioned in the figure legends, captions, and supplementary file. Data points are presented with median and interquartile ranges unless otherwise noted.

Intracellular recordings were temporally smoothed using the *smooth* function in MATLAB and the 'loess' method with span 7 at a 20.000 Hz sampling rate. Spike thresholds were set manually in either the recording or the derivative of the recording to detect spikes. Accurate spike detection was monitored manually. For the analysis of changes in membrane potential ($V_m$), the median of intracellular traces was used. Spike density of individual neurons across the 10 different trials of the short stimulation protocol was computed by using 250 ms time bins.

## Immunohistochemistry and image acquisition

For immunolabeling, respective driver lines were crossed with a *UAS-GFP-P10* or *UAS-myr-GFP* reporter to express GFP in the neurons of interest. Flies were anesthetized on ice and then transferred to 1.5 ml 4% paraformaldehyde in 0.1 M phosphate buffered saline containing 0.5% Triton- X100 (PBT). The flies were fixed in this solution for 3–6 h on a shaker followed by three 15 min wash steps in PBT. Individual fly brains were then dissected in a SYLGARD dish after immersing the fly briefly in 70% ethanol. Dissected brains were then washed three times in PBT for 15 min followed by blocking in 10% normal goat serum in 0.1 M PBT (blocking buffer), for either 2 h at room temperature or overnight at 4 °C. Next, the samples were incubated for 2 d at 4 °C in primary antibody solutions in a blocking buffer. We used rabbit anti-DILP2 diluted to 1:2000 (RRID:AB_2569969, kindly provided by J. A. Veenstra, Bordeaux, FR) to label IPCs, mouse anti-nc82 (Bruchpilot C-terminal aa 1227–1740) (*Wagh et al., 2006*) diluted as 1:500 to label the neuropils as background, and chicken anti-GFP (ab13970, abcam, Berlin, DE) diluted to 1:1000 to enhance the endogenous GFP signal. Afterward, brains were washed in PBT for three consecutive 15 min washes, followed by overnight incubation in secondary antibody solutions at 4 °C in the dark. Here, we used AlexaFluor-488 (goat anti-chicken IgY (H+L), Thermo Fisher Scientific, Waltham, MA, USA) diluted to 1:200, AlexaFluor 635 (goat anti-mouse IgG (H+L), Thermo Fisher Scientific) diluted to 1:400, and AlexaFluor 555 (goat anti-rabbit IgG (H+L), Thermo Fisher Scientific) diluted to 1:200. Finally, samples were washed three times in PBT before mounting them in Vectashield Antifade Mounting Medium (H-1000, Vector Laboratories, Newark, CA, USA).

Image acquisition was performed with a confocal laser-scanning microscope (Leica TCS SP8 WLL, Leica Microsystems, Wetzlar, DE) through the Leica Application Suite (LAS X, Leica Microsystems) using HC PL APO 10x/0.4, HC PL APO 20x/0.75 IMM, and HC PL APO 63x/1.2 CORR objectives. Fluorescence was detected using suitable lasers with a resolution of 1024×1024 pixels, in serial stacks. Final image processing to adjust contrast and brightness was done in Fiji (*Schindelin et al., 2012*).

## Survival assay

For conducting the survival assay, two replicates of 20 flies per condition were used. Initially, 3–4 d old adult female flies were wet-starved for 24 h, and then refed on either HG, HP, or SD. They were kept on these diets for the rest of their survival span and were transferred onto fresh food every second day. The number of survivors was counted once per day and dead flies were removed. For the analysis of survival percentage, we combined the counts from two replicates.

## Free-walking assay

Virgin female flies carrying *20x-UAS-Cs-Chrimson-m-venus* (*Klapoetke et al., 2014*) were mated with males from the neuromodulator GAL4 lines studied. Post eclosion, the offsprings were reared on standard food coated with 300 µM all-trans-retinal for a period of 3 to 6 d prior to the experiment. In the experiments that involved starved flies, flies were transferred to a plastic vial containing a small piece of wet tissue paper for 24 h before the start of the experiment. The walking activity of up to 20 flies was recorded using the 'UFO' behavioral setup which is based on an earlier study (*Chockley et al., 2022*). The UFO features a walking arena consisting of an inverted petri dish of 100 mm diameter as its base. To prevent escape, this base was covered with a watch glass of 120 mm diameter. The inside of the watch glass was coated with SigmaCote siliconizing reagent (cat. no. SL2; RRID:SCR_008988, Sigma-Aldrich, St. Louis, MO, USA) to prevent flies from walking upside down on the watch glass. The arena was illuminated by a ring of 60 infrared (IR) LEDs (wavelength: 870 nm) arranged concentrically around the arena. A diffuser was placed along the inside of the IR LED ring to ensure uniform illumination across the entire arena. The arena was filmed from below via a surface mirror positioned at a 45° angle below the arena and a camera (Basler acA1300-200 µm, Basler AG, Ahrensburg, DE) with a spatial resolution of 0.08 mm/pixel and a temporal resolution of 20 Hz. The camera's lens was equipped with an IR-pass filter (lower cut-off frequency: 760 nm) that only permits the IR light to pass through to ensure a strong contrast between the background (black) and flies (bright). A second ring of 60 red LEDs (wavelength: 625 nm) positioned above the IR LED ring provided the light necessary to transiently activate CsChrimson for optogenetic activation. The entire UFO setup was enclosed within a box, together with a second UFO. The box included a visible white light source placed above the arenas for controlled background light conditions.

Before each experiment, twenty flies were transferred to an empty vial and anesthetized on ice for 2 min. Subsequently, flies were transferred to the UFO arena. They were given a recovery period of 10 min to explore the arena before starting the experiment. One complete experiment consisted of five activation cycles and each activation cycle lasted 15 min and included 5 min of exposure to red light at an intensity of 3.19 mW/cm$^2$, for a total duration of 90 min per experiment. All experiments were conducted under low brightness (0.112 mW/cm$^2$) white light conditions. For each neuromodulator line, the experimental flies and control flies were recorded simultaneously in two separate UFOs situated in the same enclosure. Both the protocol and video acquisition were controlled with custom-written MATLAB code.

Movement of flies was tracked using the Caltech Fly Tracker software (*Eyjolfsdottir et al., 2014*). A detailed description of the tracking software is available at https://kristinbranson.github.io/FlyTracker/index.html. The position of individual flies was extracted from the videos on a frame-by-frame basis and features like translational and angular velocity were further computed in MATLAB. A 2 s median filter, which corresponds to less than 1% of the 300 s stimulation window, was applied prior to analysis. We excluded flies that died during the experiments from the tracking dataset prior to analysis. We conducted a minimum of two replicates for each experimental line and its corresponding controls and combined their data for further analysis. For plotting the average velocity across all flies, we implemented a data down-sampling by a factor of 20. Statistical analysis was conducted using MATLAB with the Wilcoxon rank-sum test.

## Acknowledgements

We thank Haluk Lacin (University of Missouri-Kansas City) who provided fly lines for activation experiments and Jan A Veenstra (University of Bordeaux) for sharing the DILP2 antibody. We thank Konrad Öchsner for technical assistance, Charlotte Helfrich-Förster and Wolfgang Rössler (all Julius-Maximilians-Universität of Würzburg) for sharing resources, and Christian Wegener and Meet Zandawala (both JMU) for helpful discussions. Tanja A Godenschwege (Florida Atlantic University), Chris J Dallmann, Sander Liessem, and Martina Held (all JMU) shared valuable feedback on the manuscript. This work was supported by a grant from the Deutsche Forschungsgemeinschaft (DFG) to JMA via the Emmy Noether program (DFG AC 371/1–1), and by a grant from the DFG to JMA as part of the NSF/CIHR/DFG/FRQ/UKRI-MRC Next Generation Networks for Neuroscience (Neuronex) Program (DFG AC 371/2–1).

# Additional information

## Funding

| Funder | Grant reference number | Author |
|---|---|---|
| Deutsche Forschungsgemeinschaft | DFG AC 371/1 | Jan M Ache |
| Deutsche Forschungsgemeinschaft | DFG AC 371/2 | Jan M Ache |

The funders had no role in study design, data collection and interpretation, or the decision to submit the work for publication.

## Author contributions

Rituja S Bisen, Conceptualization, Data curation, Formal analysis, Investigation, Methodology, Validation, Visualization, Writing – original draft, Writing – review and editing; Fathima Mukthar Iqbal, Formal analysis, Investigation; Federico Cascino-Milani, Formal analysis, Software, Visualization; Till Bockemühl, Methodology, Software; Jan M Ache, Conceptualization, Formal analysis, Funding acquisition, Methodology, Project administration, Resources, Software, Supervision, Validation, Visualization, Writing – original draft, Writing – review and editing

## Author ORCIDs

Rituja S Bisen ⬡ https://orcid.org/0000-0002-8312-7191
Jan M Ache ⬡ https://orcid.org/0000-0001-7355-7860

Reviewer #1 (Public review): https://doi.org/10.7554/eLife.98514.3.sa1
Reviewer #2 (Public review): https://doi.org/10.7554/eLife.98514.3.sa2
Reviewer #3 (Public review): https://doi.org/10.7554/eLife.98514.3.sa3
Author response https://doi.org/10.7554/eLife.98514.3.sa4

# Additional files

## Supplementary files

Supplementary file 1. Tables listing the p-values for statistical tests performed for data sets shown in the main figures, along with the specific numbers of individual neurons and flies analyzed for each experiment.

MDAR checklist

## Data availability

The raw data contained in all figures can be accessed on figshare.

The following dataset was generated:

| Author(s) | Year | Dataset title | Dataset URL | Database and Identifier |
|---|---|---|---|---|
| Bisen RS, Iqbal FM, Cascino-Milani F, Bockemühl T, Ache JM | 2024 | Data for: Nutritional state-dependent modulation of Insulin-Producing Cells in *Drosophila* | https://doi.org/10.6084/m9.figshare.27153867.v1 | figshare, 10.6084/m9.figshare.27153867.v1 |

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
