## [Editor Report · eLife Assessment]

With **compelling** electrophysiological and behavioural evidence, this work establishes that the activity of insulin-producing cells (IPCs) depends on the nutritional state in *Drosophila* and that, like in mammals, there is also an incretin-like effect with IPCs responding to glucose feeding but not to glucose perfusion. Moreover, the authors demonstrate that DH44 neurons respond to glucose perfusion and, together with IPCs, modulate locomotor activity. This **important** study on the neuronal regulation of metabolic homeostasis will be of interest to both neuroscience and to medical research in diabetes.

---

## [Referee Report · Reviewer #1 (Public review)]

Summary:

This study presents useful insights into the in vivo dynamics of insulin-producing cells (IPCs), key cells regulating energy homeostasis across the animal kingdom. The authors further provide compelling evidence using adult *Drosophila melanogaster* that IPCs, unlike neighboring DH44 cells, do not respond to glucose directly, but that glucose can indirectly regulate IPC activity after ingestion supporting an incretin-like mechanism in flies similar to mammals. The authors link decreased activity of IPCs to hyperactivity observed in starved flies, a locomotive behavior aimed to increase food search. Furthermore, the authors provide evidence that IPCs receive inhibitory inputs from Dh44 neurons, which are linked to increased locomotor activity.

This paper is of outstanding interest to scientists aiming to understand metabolic control of circuit dynamics, in particular for internal state-linked behaviors competing with the feeding state.

Strengths:

(1) By using whole cell patch clamp recording, the authors convincingly showed the activity pattern and regulation of IPCs and neighboring DH44 neurons under different feeding states and in various refeeding paradigms.

(2) The paper provides compelling evidence that IPCs are not directly and acutely activated by glucose, but rather through a post-ingestive incretin-like mechanism. In addition, the authors show that Dh44 neurons located adjacent to the IPCs respond to bath application of nutritive sugars contrary to the IPCs.

(3) The paper also provides useful data on the regulation of IPC activity by Dh44 neurons, which is useful to understand their regulation in vivo.

No major weaknesses remain in the revised version of this work.

---

## [Referee Report · Reviewer #2 (Public review)]

Summary

In this study, Bisen et al. characterized the state-dependency of insulin-producing cells in the brain of *Drosophila melanogaster*. They successfully established that IPC activity is modulated by the nutritional state and age of the animal. Interestingly, they demonstrate that IPCs respond to the ingestion of glucose, rather than to perfusion with it, an observation reminiscent of the incretin effect in mammals. The study is well conducted and presented and the experimental data convincingly support the claims made.

Strengths

The study makes great use of the tools available in **Drosophila** research, demonstrating the effect that starvation and subsequent refeeding have on the physiological activity of IPCs as well as on the behavior of flies to then establish causal links by making use of optogenetic tools.

It is particularly nice to see how the authors put their findings in context to published research and use for example TDC2 neuron activation or DH44 activity to establish baselines to relate their data to.

---

## [Referee Report · Reviewer #3 (Public review)]

Although insulin release is essential in the control of metabolism, adjusted to nutritional state, and plays major roles in normal brain function as well as in aging and disease, our knowledge about the activity of insulin-producing (and releasing) cells (IPCs) in vivo in limited.

In this technically demanding study, IPC activity is studied in the *Drosophila* model system by fine in vivo patch clamp recordings with parallel behavioral analyses and various optogenetic as well as feeding manipulations.

The data provide compelling evidence that IPC activity is increased with a slow time course after feeding a high glucose diet. By contrast, IPC activity is not directly affected by rising blood glucose levels. This is reminiscent of the incretin effect known from vertebrates and points to a conserved mechanism in insulin production and release upon sugar feeding.

Moreover, the data confirm earlier studies that nutritional state strongly affects locomotion. Surprisingly, strong evidence shows that IPC activity makes only a negligible contribution to this. Instead, other modulatory neurons that are directly sensitive to blood glucose levels strongly affect locomotion. Together, these data reveal a network of multiple parallel and interacting neuronal layers to orchestrate the physiological, metabolic, and behavioral responses to the nutritional state. Together with the data from a previous study, this work sets the stage to dissect the architecture and function of this network.

Strengths:

State-of-the-art current clamp in situ patch clamp recordings in behaving animals are a demanding but powerful method to provide novel insight into the interplay of nutritional state, IPC activity, and locomotion. The patch clamp recordings and the parallel behavioral analyses are of high quality, as are the optogenetic manipulations. The data showing that starvation silences IPC activity in young flies (younger than 1 week) are excellent. The evidence for the claim that locomotor activity is not increased upon IPC activity but upon the activity of other blood glucose sensitive modulatory neurons (Dh44) is compelling, too. The study provides a great system to experimentally dissect the interplay of insulin production and release with metabolism, physiology, nutritional state, and behavior. Demonstrating the incretin effect in *Drosophila* provides novel experimental routes to further study it. During the revision process, compelling evidence has been added to underscore the incretin effect, the finding that IPCs themselves do not sense sugars, and that feeding a high sugar diet does not cause unspecific stress responses.

I found no more weaknesses: The authors have carefully addressed all of my previous critiques by adding compelling new data and carefully revising the text. This paper provides a prime example of how responsible authors can utilize this constructive (but relatively new) reviewing procedure to make a very good manuscript even better.

---

## [Author Response]

The following is the authors’ response to the original reviews.

**Public Reviews:**

**Reviewer #1 (Public Review):**
Summary:This study presents useful insights into the in vivo dynamics of insulin-producing cells (IPCs), key cells regulating energy homeostasis across the animal kingdom. The authors provide compelling evidence using adult *Drosophila melanogaster* that IPCs, unlike neighboring DH44 cells, do not respond to glucose directly, but that glucose can indirectly regulate IPC activity after ingestion supporting an incretin-like mechanism in flies, similar to mammals. The authors link the decreased activity of IPCs to hyperactivity observed in starved flies, a locomotive behavior aimed at increasing food search.Furthermore, there is supporting evidence in the paper that IPCs receive inhibitory inputs from Dh44 neurons, which are linked to increased locomotor activity. However, although the electrophysiological data underlying the dynamics of IPCs in vivo is compelling, the link between IPCs and other potential elements of the circuitry (e.g. octopaminergic neurons) regulating locomotive behaviors is not clear and would benefit from more rigorous approaches.This paper is of interest to cell biologists and electrophysiologists, and in particular to scientists aiming to understand circuit dynamics pertaining to internal state-linked behaviors competing with the feeding state, shown here to be primarily controlled by the IPCs.Strengths:(1) By using whole-cell patch clamp recording, the authors convincingly showed the activity pattern of IPCs and neighboring DH44 neurons under different feeding states.(2) The paper provides compelling evidence that IPCs are not directly and acutely activated by glucose, but rather through a post-ingestive incretin-like mechanism. In addition, the authors show that Dh44 neurons located adjacent to the IPCs respond to bath application of glucose contrary to the IPCs.(3) The paper provides useful data on the firing pattern of 2 key cell populations regulating foodrelated brain function and behavior, IPCs and Dh44 neurons, results which are useful to understand their in vivo function.Weaknesses:(1) The term nutritional state generally refers to the nutrients which are beneficial to the animal. In Figure 1, the authors showed that IPCs respond to glucose but not proteins. To validate the term nutritional state the authors could test the effect of a non-nutritive sugar (e.g. D-arabinose or L-Glucose) on the post-ingestive physiological responses of the IPCs.

We thank the referee for this insightful comment. Following their suggestion, we included two new experimental data sets, which we added to Figure 1: We show that IPCs do not respond to the non-nutritive sugar D-arabinose (Figure 1H). In order to further expand this data set and our conclusions, we additionally show that IPCs do respond to fructose – a second nutritive sugar in addition to glucose (Figure 1H). Together, these data sets permit the conclusion that IPCs are sensitive to the ingestion of nutritive sugars, and do not respond to ingestion of nonnutritive sugars or high protein diets. Thus, we validate the term nutritional state.

(2) It is difficult to grasp the main message from the figures in the result section as some figures have several results subsections referring to different points the authors want to make. The key results of a figure will be easier to understand if they are summarized in one section of the results. Alternatively, a figure can be split into 2 figures if there are several key messages in those figures, e.g. Figures 2 and 3.

We appreciate this suggestion and have made several changes to our manuscript to add more clarity. Among other things, we have changed the order of data presentation in Figure 2, as suggested by the referee below, where we now start with the IPC activation data rather than the OAN activation. We also swapped the order of data presentation and split Figure S1 into Figures S1 & S2. Moreover, we re-arranged the panel order in supplementary figure S4. This significantly improved the flow of the results section. Since the figures the referee refers to contain comparative data, for example between diets (Figure 1) or neuron types (Figure 2), we prefer to keep these data sets together. However, we have carefully revised the results section to more clearly relate our statements to individual figure panels.

(3) The prime investigation of the paper is about the physiological response and locomotive behavioral readout linked to IPCs. The authors do not show a link between OANs and IPCs in terms of functional or behavioral readouts. In Figure 2 the authors first start with stating a link between OAN neurons and locomotion changes resulting from internal feeding states. The flow of the paper would be better if the authors focused on the effect of optogenetic activation of IPCs under different feeding states and their impact on fly locomotion. If the experiments done on optogenetic activation of OANs were to validate the experimental approach the data on OAN neurons is better suited for the supplement without the need of a subsection in the result section on the OANs.

We agree with the reviewer’s suggestion and switched the order of the figure panels and text to aid the flow of the manuscript. We now show and discuss the IPC activation data first (Figure 2C-H) and OAN activation afterwards (Figure 2I-K). We did keep the OAN data in the main document, though, since that facilitates comparisons between the small effects of IPC activation and the large, well-established effects of OAN activation.

(4) Figure 2F shows that optogenetic activation of IPCs in fed flies does not influence their locomotor output. In the text, the conclusion linked to Figure 2F-H states that IPC activation reduces starvation-induced hyperactivity which is a statement more suited to Figure 2I-K.

We edited the text accordingly.

(5) The authors show activation of Dh44 neurons leads to hyperpolarisation of the IPCs. What is the functional link between non-PI Dh44 neurons and the IPCs? Do IPCs express DH44R or is DH44 required for this effect on IPCs? Investigating a potential synaptic or peptidergic link between DH44 neurons and IPCs and its effect on behavior would benefit the paper, as it is so far not well connected.

Although we have not performed any experiments dedicated to investigating the functional link between DH44Ns outside the PI and the IPCs in this study, there are two lines of evidence supporting that this connection is relatively direct. First, IPCs do express DH44R1 & R2, as we show in a parallel study in eLife (Held M, et al. ‘Aminergic and peptidergic modulation of Insulin-Producing Cells in *Drosophila*’. eLife. 2024;13. doi:10.7554/ELIFE.99548.1). Second, we performed functional connectivity experiments using a Leucokinin (LK) driver line in that paper. This driver line labels two pairs of non-PI DH44Ns in the VNC, which are DH44 and LK positive (Zandawala et al 2018). Activating that line leads to inhibition of IPCs, similar to the effect we observed here for DH44N activation. These two lines of evidence suggest that there could be a direct peptidergic connection between DH44+ neurons and IPCs. We have added a paragraph mentioning these experiments to our discussion:

‘Notably, the DH44^PI^Ns express the DH44 peptide, as confirmed by anti-DH44 stainings(100). This also applies to a large fraction of neurons labelled in the broad DH44 driver line(100). However, a subset of neurons labelled in the broad line did not exhibit DH44 immunoreactivity(100), and might therefore not actually express the DH44 peptide. Hence, the inhibition of IPCs could be driven by neurons in the DH44 driver line that do not express DH44. A strong candidate for the inhibition are LK and DH44-positive neurons, which are labelled by the broad line(76). In a parallel study, we showed that LK-expressing neurons strongly inhibit IPCs(30), similar to the broad DH44 line used here. Furthermore, evidence from single-nucleus transcriptomic analysis shows that IPCs express DH44-R1 and DH44-R2 receptors(30). Therefore, it is possible that DH44Ns communicate with IPCs through a direct peptidergic connection. Notably, the inhibitory effect of non-PI DH44Ns on IPCs was very strong and fast, suggesting that a connection via classical synapses is more likely. Regardless, our results show that the glucose sensing DH44^PI^Ns and IPCs act independently of each other.’

**Reviewer #2 (Public Review):**
Summary:In this study, Bisen et al. characterized the state-dependency of insulin-producing cells in the brain of **Drosophila melanogaster**. They successfully established that IPC activity is modulated by the nutritional state and age of the animal. Interestingly, they demonstrate that IPCs respond to the ingestion of glucose, rather than to perfusion with it, an observation reminiscent of the incretin effect in mammals. The study is well conducted and presented and the experimental data convincingly support the claims made.Strengths:The study makes great use of the tools available in **Drosophila** research, demonstrating the effect that starvation and subsequent refeeding have on the physiological activity of IPCs as well as on the behavior of flies to then establish causal links by making use of optogenetic tools.It is particularly nice to see how the authors put their findings in context to published research and use for example TDC2 neuron activation or DH44 activity to establish baselines to relate their data to.Weaknesses:I find the inability of SD to rescue the IPC starvation effect in Figure 1G&H surprising, given that the fully fed flies were raised and kept on that exact diet. Did the authors try to refeed flies with SD for longer than 24 hours? I understand that at some point the age effect would also kick in and counteract potential IPC activity rescue. I think the manuscript would benefit if the authors could indicate the exact age of the SD refed flies and expand a bit on the discussion of that point.

We have expanded the first paragraph of our discussion to tackle these questions, in particular the potential effect of aging, as suggested by the referee. We now also indicate the exact age of the flies. Moreover, we have conducted additional experiments in which we added either glucose or arabinose to our standard diet (Figure 1H). As we would have expected based on our hypothesis that the glucose concentration in our standard diet was too low to cause an increase in IPC activity after starvation, we find that feeding standard diet plus glucose increases IPC activity to the same level as glucose only, and that adding arabinose to the standard diet does not lead to increased IPC activity after starvation (Figure 1H).

The incretin-like effect is exciting and it will be interesting in the future to find out what might be the signal mediating this effect. It is interesting that IPCs in explants seem to be responsive to glucose. I think it would help if the authors could briefly discuss possible sources for the different findings between these in fact very different preparations. Could the the absence of the inhibitory DH44 feedback in the *ex-vivo* recordings for example play a role?

We thank the referee for this interesting point and expanded our discussion accordingly. We included that, in particular in brain explants without a VNC, the inhibitory connection we describe might be absent, as the referee suggested: ‘Previous ex vivo studies suggested that IPCs, like pancreatic beta cells, sense glucose cell-autonomously(23,24). Consistent with this, we observed an increase in IPC activity after the ingestion of glucose (Figure 2B). However, IPC activity did not increase during the perfusion of glucose directly over the brain. Importantly, the fly preparations were kept alive for several hours allowing the glucose-rich saline to enter circulation and reach all body parts. Several factors may explain the difference between ex vivo and in vivo preparations. First, in ex vivo studies, certain regulatory feedback mechanisms present in vivo could be absent. For example, the strong inhibitory input IPCs receive from DH44Ns we found would likely be absent in brain explants without a VNC. A lack of inhibitory feedback might allow for more direct glucose sensing by IPCs ex vivo, whereas in vivo, the IPC response could be suppressed by more complex systemic feedback. Second, we attempted to use the intracellular saline formulation employed in a previous ex vivo study44. However, we observed that IPCs depolarized quickly using this saline, leading to unstable recordings that did not meet our quality standards for in vivo experiments. Another possible explanation for the lack of an effect of glucose might have been that the dominant circulating sugar in flies is trehalose(70,71) which is derived from glucose. When we extended our experiments, we found that trehalose perfusion did not affect IPC activity either, strengthening the idea that IPCs do not directly sense changes in hemolymph sugar levels. Therefore, our findings suggest that, similar to mammals, IPC activity and hence, insulin release, is not simply modulated by hemolymph sugar concentration in *Drosophila*.’

The incretin-like effect the authors observed seems to start only after 5h which seems longer than in mammals where, as far as I know, insulin peaks around 1h. Do the authors have ideas on how this timescale relates to ingestion and glucose dynamics in flies?

We have now included the following section in the discussion to explicitly address the question of different activity dynamics in flies and mammals, but also the limitations of our electrophysiological approach in this regard: ‘We observed that IPC activity increased over a timescale of hours, which is longer compared to the fast insulin response in mammals, where insulin typically peaks within an hour of feeding(97). In flies, insulin levels rise within minutes of refeeding, followed by a drop after 30 min(20). Our experimental techniques limit our ability to capture these fast initial dynamics, since the preparation for intracellular recordings requires tens of minutes, so that we typically recorded IPC activity at least 20 min after the last food ingestion. Notably, studies in fasted mammals have shown that insulin peaks within minutes of refeeding, followed by a rapid decline, with levels stabilizing as feeding continues(98,99). We speculate a similar dynamic could be present in flies, but with our approach, we capture the steady-state reached tens of minutes after food ingestion rather than a potential initial peak.’

The authors mention "a decrease in the FV of IPC-activated starved flies even before the first optogenetic stimulation (Figure 2I),". Could this be addressed by running an experiment in darkness, only using the IR illumination of their behavioral assay?

We thank the referee for pointing out this unexpected result. We discuss this in more detail in the new version of our manuscript and expand on the reasons for not performing these optogenetic activation experiments in the dark: First, the red LED required to activate CsChrimson triggers strong startle responses in dark-adapted flies, which mask other behavioral effects, in particular subtle ones such as those observed for IPCs. The startle response is much reduced when performing experiments under low background light conditions. Second, flies, at least in our hands, do not exhibit robust foraging behavior or starvation-induced hyperactivity in the dark, which is critical for our behavioral experiments. However, we also explain in our discussion that we believe the effect of background illumination is relatively small, since flies expressing CsChrimson in OANs or DH44Ns show comparable activity levels to controls. Hence, a part of this effect is likely attributable to leak currents induced by CsChrimson expression. We would like to point out though that we are careful in our description of the IPC effect on behavior, and focus on the fact that it is considerably smaller than the effects of other modulatory neurons (DH44Ns and OANs).

The authors show an inhibitory effect of DH44 neuron activation on IPC activity. They further demonstrate that DH44PI neurons are not the ones driving this and thus conclude that "...IPCs are inhibited by DH44Ns outside the PI.". As the authors mentioned the broad expression of the DH44-Gal4 line, can they be sure that the cells labeled outside the PI are actually DH44+? If so they should state this more clearly, if not they should adapt the discussion accordingly.

We have substantially added to our discussion of this point, according to the referee’s great suggestion. In short, the broad line includes neurons that are DH44 positive and neurons that are not: ‘Notably, the DH44^PI^Ns express the DH44 peptide, as confirmed by anti-DH44 stainings(100). This also applies to a large fraction of neurons labelled in the broad DH44 driver line(100). However, a subset of neurons labelled in the broad line did not exhibit DH44 immunoreactivity(100), and might therefore not actually express the DH44 peptide. Hence, the inhibition of IPCs could be driven by neurons in the DH44 driver line that do not express DH44.’

**Reviewer #3 (Public Review):**
Although insulin release is essential in the control of metabolism, adjusted to nutritional state, and plays major roles in normal brain function as well as in aging and disease, our knowledge about the activity of insulin-producing (and releasing) cells (IPCs) in vivo is limited.In this technically demanding study, IPC activity is studied in the *Drosophila* model system by fine in vivo patch clamp recordings with parallel behavioral analyses and optogenetic manipulation.The data indicate that IPC activity is increased with a slow time course after feeding a high-glucose diet. By contrast, IPC activity is not directly affected by increasing blood glucose levels. This is reminiscent of the incretin effect known from vertebrates and points to a conserved mechanism in insulin production and release upon sugar feeding.Moreover, the data confirm earlier studies that nutritional state strongly affects locomotion. Surprisingly, IPC activity makes only a negligible contribution to this. Instead, other modulatory neurons that are directly sensitive to blood glucose levels strongly affect modulation. Together, these data indicate a network of multiple parallel and interacting neuronal layers to orchestrate the physiological, metabolic, and behavioral responses to nutritional state. Together with the data from a previous study, this work sets the stage to dissect the architecture and function of this network.Strengths:State-of-the-art current clamp in situ patch clamp recordings in behaving animals are a demanding but powerful method to provide novel insight into the interplay of nutritional state, IPC activity, and locomotion. The patch clamp recordings and the parallel behavioral analyses are of high quality, as are the optogenetic manipulations. The data showing that starvation silences IPC activity in young flies (younger than 1 week) are compelling. The evidence for the claim that locomotor activity is not increased upon IPC activity but upon the activity of other blood glucose-sensitive modulatory neurons (Dh44) is strong. The study provides a great system to experimentally dissect the interplay of insulin production and release with metabolism, physiology, and behavior.Weaknesses:Neither the mechanisms underlying the incretin effect, nor the network to orchestrate physiological, metabolic, and behavioral responses to nutritional state have been fully uncovered. Without additional controls, some of the conclusions would require significant downtoning. Controls are required to exclude the possibility that IPCs sense other blood sugars than glucose. The claim that IPC activity is controlled by the nutritional state would require that starvation-induced IPC silencing in young animals can be recovered by feeding a normal diet. At current firing in starvation, silenced IPCs can only be induced by feeding a high-glucose diet that lacks other important ingredients and reduces vitality. Therefore, feasible controls are needed to exclude that diet-induced increases in IPC firing rate are caused by stress rather than nutritional changes in normal ranges. The finding that refeeding starved flies with a standard diet had no effect on IPC activity but a strong effect on the locomotor activity of starved flies contradicts the statement that locomotor activity is affected by the same dietary manipulations that affect IPC activity. The compelling finding that starvation induces IPC firing would benefit from determining the time course of the effect. The finding that IPCs are not active in fed animals older than 1 week is surprising and should be further validated.

We thank the referee for the thoughtful and constructive criticism of our experiments and conclusions. Below, we lay out how we tackled the individual points raised by the referee.

(1) ‘Controls are required to exclude the possibility that IPCs sense other blood sugars than glucose.’

To address this point, we conducted experiments in which we perfused trehalose (Figure 3B), the main circulating hemolymph sugar in *Drosophila* and other insects. Our results clearly show that trehalose does not affect IPC activity upon perfusion, confirming our statements that IPCs do not sense key blood sugars directly.

(2) ‘Feasible controls are needed to exclude that diet-induced increases in IPC firing rate are caused by stress rather than nutritional changes in normal ranges’.

We agree with the referee that this point was not completely fleshed out in our first submission. We have now performed additional experiments in which we added glucose (and fructose) to our standard diet (Figure 1H). Flies feeding on this diet received all necessary nutrients but still experienced high concentrations of sugars. The effects of high glucose in a standard diet background were indistinguishable from those of high glucose in agarose, confirming that the IPCs respond to sugar rather than stress. Another important observation in this context is that IPCs in flies kept on a high protein diet exhibited much lower spike rates than flies exhibiting the high glucose diet, even though they had a much shorter lifespan and therefore, presumably, experienced much higher stress levels (Figure 1H, Figure S1). These observations underline that stress is certainly not the primary factor here.

(3) ‘The finding that refeeding starved flies with a standard diet had no effect on IPC activity but a strong effect on the locomotor activity of starved flies contradicts the statement that locomotor activity is affected by the same dietary manipulations that affect IPC activity.’

We have revised the respective section of the results and discussion accordingly and are more careful and clearer in our interpretation of this behavioral dataset: ‘These results show that the locomotor activity was affected by the same dietary manipulations that had strong effects on IPC activity. However, IPC activity changes alone cannot explain the modulation of starvation-induced hyperactivity. On the one hand, high-glucose diets which drove the highest activity in IPCs were not sufficient to reduce locomotor activity back to baseline levels. On the other hand, refeeding flies with SD did not revert the effects of starvation on IPC activity (Figure 1H), but it was sufficient to reduce the locomotor activity below baseline levels (Figure 2B). This suggests that the modulation of starvation-induced hyperactivity is achieved by multiple modulatory systems acting in parallel.’

(4) ‘The compelling finding that starvation induces IPC firing would benefit from determining the time course of the effect.’

We followed the referee’s excellent suggestion and determined the time course of the starvation effect in three timesteps, similar to the experiments we did for refeeding (Figure 1G). In addition, we now also quantify the number of active IPCs (i.e., IPCs that fired at least one action potential during our five-minute analysis window), which further illustrates the dynamics of the starvation and refeeding effects. We find that the starvation effect is graded, and that IPC activity decreases with increasing starvation duration.

(5) ‘The finding that IPCs are not active in fed animals older than 1 week is surprising and should be further validated.’

To address the referee’s comment, we have added 14 new IPC recordings from flies in the 6–26-day range, such that we now have recordings from 9-14 IPCs for each age range (Figure S2B). They confirmed our previous analysis and strengthened the finding that IPC activity dramatically decreases after 8 days (on our standard diet). The total number of IPCs in this supplementary dataset was thus increased from 34 to 48.

**Recommendations for the authors:**

**Reviewer #1 (Recommendations For The Authors):**
(1) Do IPCs respond to glucose specifically after ingestion or generally to any other nutritive sugars? To tackle this question the IPC responses in starved flies can be recorded after refeeding flies with other nutritive sugars (fructose, sucrose).

To address this important question, we have performed additional experiments in which we refed starved flies with fructose, as a nutritive sugar, and arabinose, as a non-nutritive sugar. As expected, IPCs responded to fructose but not arabinose and hence nutritive sugars in general. We describe and discuss these key results in the new version of our manuscript.

(2) In Figure 2, the x and y axes are not annotated on all subfigures, which might help improve clarity.

We have annotated the subfigures as requested.

(3) In the discussion on page 9 ("...we observed an increase in IPC activity after the ingestion of glucose (Figure 2B)."), the authors refer to Figure 2B instead of 3C.

We have fixed this oversight.

**Reviewer #2 (Recommendations For The Authors):**
IntroductionI think it could be helpful for the reader if you would briefly state the number of IPCs and whether you are targeting all of them with Dilp2-Gal4.

We included the numbers according to the suggestion. 14 IPCs are labeled in the driver line, and this is the number of IPCs commonly assumed to be present in the PI.

FiguresIn some Figures (for example 1D & E) the authors state the number of IPCs recorded (N) but not the number of animals used (n). This should be stated as the data from within an animal are dependent and might give insights about IPC heterogeneity.

We have compiled tables for the supplementary material (Tables S5 & S6) in which we state the number of IPCs and DH44^PI^Ns recorded and the number of different flies for each figure panel. We have recorded an average of 1.4 IPCs per fly (217 IPCs from 160 flies). We therefore expect the bias introduced by individual flies to be rather small. However, in our parallel study, we specifically investigate the heterogeneity of IPCs by maximizing the number of IPCs recorded per fly (Held M, et al. ‘Aminergic and peptidergic modulation of Insulin-Producing Cells in *Drosophila*’. eLife. 2024;13. doi:10.7554/ELIFE.99548.1). In the case of DH44PINs, we recorded 24 neurons in 21 flies – 1.1 neurons per fly.

- Figure 3D: There is some white visible among the cell bodies in the overlay. I assume this comes from projecting across layers rather than indicating DH44 - IPC overlap? It would help to explicitly state that.

We have added a statement to the results section, in which we explain that most of the white is due to overlap in the z-projection rather than overlap in the driver lines. However, there are few cases (typically one to two cells per brain), in which neurons labeled by the DH44 line also stain positive for Dilp2, indicating they express both neuropeptides. We have added this information to the manuscript:

Results: ‘DH44^PI^Ns are anatomically similar to IPCs, and their cell bodies are located directly adjacent to those of IPCs in the PI, making them an ideal positive control for our experiments (Figure 3D). A small subset of DH44^PI^Ns also expresses Dilp2(75), and our immunostainings confirmed colocalization of Dilp2 and DH44 in a single neuron (Figure 3D, white arrow).’

In figure caption: ‘UAS-myr-GFP was expressed under a DH44-GAL4 driver to label DH44 neurons. GFP was enhanced with anti-GFP (green), brain neuropils were stained with anti-nc82 (cyan), and IPCs were labelled using a Dilp2 antibody (magenta). White arrow indicates Dilp2 and DH44-GAL4 positive neuron. The other white regions in the image result from an overlap in z-projections between the two channels, rather than from antibody colocalization.’

- Figure 4I: One might get the impression that the fast onset peak of activity precedes the stimulation onset, using a thinner line width might help avoid that.

This effect is due to a combination of using relatively heavy lines for clear visibility of the data and a gentle smoothing step (a 2s median filter, which corresponds to less than 1% of the 300s stimulation window) in our analysis of the behavioral data. However, inspection of the raw data clearly shows increases in velocity after the onset of the optogenetic activation. We clarified this in the figure caption: ‘Average FV across all DH44N activation trials based on two independent replications of the experiment in I. Note that the peak in average FV lies within the first frame of the stimulation window.’

- S3 panel letters do not match references in the text.

We fixed this oversight.

Formatting- Page 10: The paragraphs on the bottom of the page got switched around.

This has been fixed.

- Page 14: The first paragraph after the header "Free-walking assay" seems to be coming from elsewhere.

We apologize for this slightly embarrassing mistake. We used our related bioRxiv preprint (Held et al.) as a template for formatting this paper, and accidentally left this part of the methods section in the manuscript. We have fixed this error in our resubmission.

**Reviewer #3 (Recommendations For The Authors):**
Major suggestions:(1) The data show convincingly that IPC activity is decreased by starvation during the first week of adult life (Figures 1C and D). However, the conclusion that IPC activity is controlled by the nutritional state requires additional care. First, refeeding starved adult animals with a normal diet does not bring back normal IPC firing rates (Figure 1H). Therefore, IPC activity does not strictly follow changes in nutritional state, but IPCs are silenced by starvation. Second, from the second week of adult life on, IPCs are silent anyway, and thus unlikely responsive to changes in the nutritional state anymore (which might be different on a different standard diet?) The only effect of feeding on IPC activity is observed upon feeding starved, young animals with high glucose for 12-24 h (Figure 1G). However, it is not clear whether increased IPC firing is caused by the effects of high glucose on the nutritional state in a normal range, or because of diet-induced stress (the diet also severely shortens lifespan, Figure 1S). Does high glucose also increase IPC firing rate in young, fed animals? These would have strongly increased glucose concentrations but not suffer the stress of not getting any other nutrients. Such experiments would be required to make the statement that glucose feeding increases IPC firing rate.

We have performed several experiments to address this criticism. First, we performed a time course analysis of the starvation effect. We show that the IPC activity reduction is graded, and that IPC activity declines already after two hours of starvation, a timepoint at which stress levels should still be relatively small (Figure 1G). Second, we refed flies with high glucose concentrations added to the standard diet (Figure 1H). This minimized any potential stress responses due to a lack in nutrients. Third, we now show that IPCs specifically respond to nutritive (glucose and fructose), but not to non-nutritive sugars (arabinose, Figure 1H). We believe that these data sets, in addition to the graded refeeding effect, make a strong case for the nutritional state dependent modulation of IPCs.

(2) The testing of locomotor activity is well done, nicely recapitulates starvation-induced increases in locomotion, and adds interesting novel findings on refeeding with high glucose versus high protein diet. However, the statement that locomotor activity was affected by the same dietary manipulations that had strong effects on IPC activity does not reflect the data presented. Refeeding starved flies with a standard diet had no effect on IPC activity (Figure 1H) but a strong effect on locomotor activity of starved flies (a strong reduction, even stronger than high glucose diet, Figure 2B).

We have revised the respective section of the results and discussion accordingly and are more careful and clearer in our interpretation of this behavioral dataset: ‘These results show that the locomotor activity was affected by the same dietary manipulations that had strong effects on IPC activity. However, IPC activity changes alone cannot explain the modulation of starvationinduced hyperactivity. On the one hand, high-glucose diets which drove the highest activity in IPCs were not sufficient to reduce locomotor activity back to baseline levels. On the other hand, refeeding flies with SD did not revert the effects of starvation on IPC activity (Figure 1H), but it was sufficient to reduce the locomotor activity below baseline levels (Figure 2B). This suggests that the modulation of starvation-induced hyperactivity is achieved by multiple modulatory systems acting in parallel.’

Related to points 1 and 2, a key statement that the results establish that IPC activity is controlled by the nutritional state requires care. What the data convincingly show is that IPC activity is near zero upon starvation.

As described above, we have added several extensive data sets (fructose feeding, arabinose feeding, trehalose perfusion, starvation time course) to show that we indeed observe a nutritional state dependent modulation of IPCs and describe these new results in the results and discussion.

(3) The time course of nutritional state-dependent changes of IPC activity is claimed to be slow, several hours to days. Unless I have missed a figure, the underlying data are not presented (only for high glucose diet). It would be great if this could also be shown for a standard diet with higher glucose concentrations than the one used so that it rescues starvation-induced IPC silencing without shortening lifespan (if this is feasible?). The data showing starvation-induced IPC silencing are convincing, but, unless I have missed it, the time course has not been determined. It would be very nice to actually show this. Have different starvation times been tested in relation to IPC firing rate, and if yes, with what time resolution? Does IPC activity change already after 0.5 or 1 or a few hours of starvation? If starvation can silence IPCs faster than assumed, the nearzero IPC activity in animals older than a week could very well be caused by longer time intervals between meals.

We have performed experiments to address both important points raised by the referee here. (1) We have added high glucose concentrations to our standard diet, and show that it has the same effect – a significant increase in IPC activity – as the high glucose diet (Figure 1H). (2) We have analyzed the time course of IPC activity reduction in response to starvation (Figure 1G). Indeed, we find that a few hours of starvation start reducing IPC activity. We discuss the possibility that reduced IPC activity in older flies could be due to reduced food intake: ‘One of our experiments demonstrated that IPC activity was heavily diminished in flies older than 10 days (Figure S2B). A possible explanation could be that flies feed less as they age. However, this only holds true for flies older than 14 days86. Therefore, reduced IPC activity in 10-11 day old flies is unlikely to result from reduced food intake and likely involves inhibition of insulin signaling.’

(4) The data on the proposed incretin effect are of high importance in potentially highlighting a highly conserved link between glucose ingestion and insulin release. An important control would be to test different sugars, such as trehalose, an important blood sugar of flies. If glucose is converted into trehalose and this is what IPCs sense, then perfusion of glucose has no effect. The fact fantastic experiments show that the DH44 neurons are sensitive to glucose perfusion does rule out that IPCs sense a different sugar. This would be very different from the incretin effect that requires additional hormones. In addition, as mentioned above, controls are required to show that high glucose affects IPCs as a nutrient and not as a stressor (see point 1), for example refeeding with a standard diet that contains a higher glucose concentration but does not reduce lifespan. Another great control to solidify the exciting claim on the incretin effect would be to knock out candidate *Drosophila* incretin hormones and test whether a high glucose diet stops increasing the IPC firing rate (although simpler controls might also do the job).

We have performed the two key experiments suggested by the referee. (1) We perfused trehalose as the primary blood sugar of flies and showed that IPCs do not respond to trehalose perfusion (Figure 3B & C). This further strengthens the finding that IPC activity in flies shows an incretin-like effect. (2) We have added high concentrations of glucose to our standard diet to provide flies with a full diet that contains high glucose concentrations. IPC activity in these flies was indistinguishable from the activity in flies which consumed pure glucose diets. In contrast, IPC activity in flies kept on a high protein diet, which dramatically reduced lifespan, was very low. These results clearly show that higher IPC activity is not due to increased stress levels, but a function of nutritive sugar ingestion. We further validated this hypothesis by refeeding flies with fructose as a nutritive sugar, which increased IPC activity, and arabinose as a non-nutritive sugar, which did not affect IPC activity (Figure 1H).

Another point that might be relevant to this discussion is that IPC activity is almost entirely shut down during flight in *Drosophila* (which we showed in Liessem et al. 2023, Current Biology 33 (3), 449-463. e5). Several ‘stress hormones’ are released during flight, including octopamine. The fact that IPC activity is low in flying flies, starved flies, and flies kept on a pure protein diet (which all experience high stress levels), to us, very clearly suggests that stress is not the predominant factor here. We would also like to point out that, while the lifespan was reduced in flies kept on pure glucose diets, survival rates were at 100% until day 14, and we carried out our experiments on day 2 after starvation. Hence, these flies might not (yet) experience particularly high stress levels.

(5) The discussion relates the absence of IPC firing in animals older than 1 week to aging. However, given that the flies fed on a normal diet show the typical lifespan for *Drosophila*, a 10-dayold fly is still in its youth. Maybe flies at 10 days eat simply less and thus IPC spiking goes down as in starved flies, especially because the standard diet used contains low glucose. Do IPCs also become silent after a week if the animals are fed with a standard diet that contains a higher glucose concentration? Without additional controls, this part of the discussion is pretty speculative and should be revised.

We agree with the reviewer, that it is not clear whether reduced IPC activity is a direct result of physiological changes that occur with aging, or an indirect effect of reduced food intake, which occur during aging. In both cases, in our view, it would be an age-related effect. Since this is a minor point of our manuscript, we decided not to perform additional experiments, other than significantly increasing the sample size for the aging data set already presented to shore up our findings (Figure S2B). We have, however, revisited the discussion of this point according to the referee’s suggestion: ‘One of our experiments demonstrated that IPC activity was heavily diminished in flies older than 10 days (Figure S2B). A possible explanation could be that flies feed less as they age. However, this only holds true for flies older than 14 days(85). Therefore, reduced IPC activity in 10-11 day old flies is unlikely to result from reduced food intake and likely involves inhibition of insulin signaling.’

Other suggestions:(6) For the mixed effects of octopamine and tyramine on larval locomotion that are referred to, it might be interesting to also look at Schützler et al 2019, PNAS because it shows that starvation activates TBH so that the octopamine to tyramine ratio is increased.

We refer to Schützler et al. in the following paragraph of our discussion: ‘This intermittent locomotor arrest has been previously described in adult flies and is thought to be mediated by ventral unpaired median OANs, which have been suggested to suppress long-distance foraging behavior(69). Since these are not the only neurons we activate in the TDC2 line, we speculate that the stopping phenotype could also result from concerted effects of octopamine and tyramine modulating muscle contractions(65-67) and motor neuron excitability(68), as previously described in *Drosophila* larvae, or from OANs interfering with pattern generating networks in the ventral nerve cord (VNC) during longer activation(69).’

(7) The reference list requires care. For example, reference 43 is identical to 67, reference 66 gives no information on incretin-like hormones in *Drosophila* as stated in the text

We carefully double-checked our reference list and corrected the mistakes mentioned.